# FUNCTION-CONSISTENT FEATURE DISTILLATION

**Dongyang Liu**[1,2]**, Meina Kan**[1,2]**, Shiguang Shan**[1,2,3]**, Xilin CHEN**[1,2]
[1] Key Lab of Intell. Info. Process., Inst. of Comput. Tech., CAS
[2] University of Chinese Academy of Sciences  [3] Peng Cheng Laboratory
`{liudongyang21s, kanmeina, sgshan, xlchen}@ict.ac.cn`

## ABSTRACT

Feature distillation makes the student mimic the intermediate features of the teacher. Nearly all existing feature-distillation methods use L2 distance or its slight variants as the distance metric between teacher and student features. However, while L2 distance is isotropic w.r.t. all dimensions, the neural network's operation on different dimensions is usually anisotropic, *i.e.*, perturbations with the same 2-norm but in different dimensions of intermediate features lead to changes in the final output with largely different magnitude. Considering this, we argue that the similarity between teacher and student features should *not* be measured merely based on their appearance (*i.e.*, L2 distance), but should, more importantly, be measured by their difference in function, namely how later layers of the network will read, decode, and process them. Therefore, we propose Function-Consistent Feature Distillation (FCFD), which explicitly optimizes the functional similarity between teacher and student features. The core idea of FCFD is to make teacher and student features not only numerically similar, but more importantly produce similar outputs when fed to the later part of the same network. With FCFD, the student mimics the teacher more faithfully and learns more from the teacher. Extensive experiments on image classification and object detection demonstrate the superiority of FCFD to existing methods. Furthermore, we can combine FCFD with many existing methods to obtain even higher accuracy. Our codes are available at https://github.com/LiuDongyang6/FCFD.

## 1 INTRODUCTION

Deep neural networks (DNNs) have demonstrated great power on a variety of tasks. However, the high performance of DNN models is often accompanied by large storage and computational costs, which barricade their deployment on edge devices. A common solution to this problem is Knowledge Distillation (KD), whose central idea is to transfer the knowledge from a strong teacher model to a compact student model, in hopes that the additional guidance could raise the performance of the student (Gou et al., 2021; Wang & Yoon, 2021). According to the type of carrier for knowledge transfer, existing KD methods can be roughly divided into the following two categories:

### 1.1 KNOWLEDGE DISTILLATION BASED ON FINAL OUTPUT

Hinton et al. (2015) first clarifies the concept of knowledge distillation. In their method, softened probability distribution output by the teacher is used as the guidance to the student. Following works (Ding et al., 2019; Wen et al., 2019) step further to explore the trade-off between soft logits and hard task label. Zhao et al. (2022) propose to decouple target-class and non-target-class knowledge transfer, and Chen et al. (2022) make the student to reuse the teacher's classifier. Park et al. (2019) claims that the relationship among representations of different samples implies important information, and they used this relationship as the knowledge carrier. Recently, some works introduce contrastive learning to knowledge distillation. Specifically, Tian et al. (2020) propose to identify if a pair of teacher and student representations are congruent (same input provided to teacher and student) or not, and SSKD (Xu et al., 2020) introduce an additional self-supervision task to the training process. Whereas all the aforementioned methods employ a fixed teacher model during distillation, Zhang et al. (2018b) propose online knowledge distillation, where both the large and the small models are randomly initialized and learn mutually from each other during training.

## 1.2 Knowledge Distillation based on Intermediate Features

Comparing with final output, intermediate features contain richer information, and extensive methods try to use them for distillation. FitNet (Romero et al., 2015) pioneers this line of works, where at certain positions, intermediate features of the student are transformed to mimic corresponding teacher features w.r.t. L2 distance. FSP (Yim et al., 2017) and ICKD (Liu et al., 2021) utilize the Gramian matrix to transfer knowledge, and AT (Zagoruyko & Komodakis, 2017) uses the attention map. Generally, one student feature only learns from one teacher feature. However, some works have extended this paradigm: Review (Chen et al., 2021b) makes a student layer not only learn from the corresponding teacher layer, but also from teacher layers prior to that; AFD (Ji et al., 2021) and SemCKD (Chen et al., 2021a) make a student layer to learn from all candidate teacher layers, with attention mechanism proposed to allocate weight to each pair of features. Considering that negative values are filtered out by ReLU activation, Heo et al. (2019) propose the marginal ReLU activation and the partial L2 loss function to suppress the transfer of useless information.

## 1.3 Our motivation

Extensive methods have been proposed to enhance feature distillation, from perspectives like distillation position (Chen et al., 2021b; Ji et al., 2021; Chen et al., 2021a), feature transformation (Yue et al., 2020; Zhang et al., 2020; Lin et al., 2022), *etc*. However, all these methods use L2 distance or its slight variants (Heo et al., 2019) as the distance function to optimize. In this paper, we argue that the simple L2 distance (as well as L1, smooth L1, *etc*.) has significant weakness that impedes itself from being an ideal feature distance metric: it measures features' similarity independently to the context. In other words, L2 distance merely measures the numerical distance between two features, which can be considered as the **appearance** of the feature. This kind of appearance-based distance treats all dimensions in an isotropic manner, without considering feature similarity in **function**. The function of intermediate features is to serve as the input to the network's later part. Features playing similar functions express similar semantics and lead to similar results. Since neural network's operation on different dimensions are usually anisotropic, changing the feature by the same distance along different dimensions can lead to changes in the final output with completely different magnitude. As a result, with only L2-based feature matching, student features with a large variety of functional differences to teacher feature are considered equally-good imitations, which inevitably impedes knowledge transfer. On the other hand, suppose that among these imitations, an additional function-based supervision could be provided to identify the better ones that lead to less change in the final output, by enforcing the student to prioritize them, the semantic consistency between teacher and student features can be better guaranteed.

As a toy example, considering a network that computes $(5x^2 - 1)^4 + 5(2x + 2)^2$. Assume that the network can be divided into 2 cascaded modules: the first calculates $(m_1, m_2) = (5x^2 - 1, 2x + 2)$, and the second calculates $out = m_1{}^4 + 5m_2{}^2$ (Fig. 1(a)). With $x = 1$ as input, we have $(m_1, m_2) = (4, 4)$ and $out = 336$. Now let a student to mimic the intermediate feature, *i.e.*, $(m_1, m_2)$. Considering two possible solutions: $(3, 4)$ and $(4, 3)$, while L2 distance tells they are equally good, the function perspective points out that $(4, 3)$ is clearly better because it leads to less deviation in final output. *Comparing with*

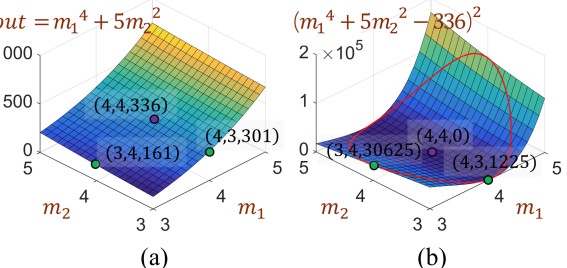

(a)                    (b)

Figure 1: A toy example illustrating the relationship between appearance and function. Points on the red ring in (b) have the same L2 distance to target (4,4), but they cause largely different deviations in final output.

$m_2$, *changes on* $m_1$ *has stronger impact on the final output, which means task-relevant information changes more rapidly along* $m_1$. *The student should thus pay more attention to* $m_1$ *for learning more task-relevant information. Unfortunately, with only L2-based feature distillation, no such impetus is available to push the student to prioritize* $(4, 3)$ *to* $(3, 4)$. As Fig. 1(b) shows, all points on the red ring have the same L2 distance to the mimicking target, while their function varies greatly. We reasonably speculate that if an extra matching mechanism, which drives the student to the solution with highest function consistency, could be provided, knowledge transfer will be more effective

and student could benefit more from distillation. The aforementioned phenomenon also exists in practical tasks and networks: with a ResNet32x4 model pre-trained on CIFAR100, after randomly selecting 512 directions and move intermediate features along these direction by certain L2 distance, the KL-Divergence between the final output before and after feature manipulation averagely ranges from 0.59 to 1.29. When always selecting directions causing the greatest difference in output, the accuracy on the whole validation set drops by 24.42%, while it only drops by $10.69\%$ when always selecting the directions causing the slightest difference. The results inspire us that even when the student has well mimicked the teacher's features w.r.t. L2 distance, there is still large room for further improvement if such guidance that differentiates **functional** differences could be provided.

In this paper, we propose Function-Consistent Feature Distillation (FCFD), which explicitly optimizes the functional similarity between intermediate features from teacher and student. We clarify that the similarity between features is not only determined by the features themselves, but more importantly defined by how the later layers will read, decode, and process them. Therefore, we introduce a novel function-based feature-matching mechanism, which takes the information about later network layers into consideration. The proposed mechanism could tell the good and bad from features that are considered equally good by L2 distance, thus overcoming the weakness of existing feature distillation methods. In this way, FCFD enables the student to mimic the teacher more faithfully and learn more from the teacher. Overall, *our main contributions are summarized as below*:

1. We propose a novel knowledge distillation mechanism named FCFD, which emphasizes the matching between teacher and student intermediate features w.r.t. function.

2. Extensive experiments on image classification and object detection are conducted to investigate the effectiveness and generalization of FCFD. The results show that FCFD significantly outperforms state-of-the-art knowledge distillation methods.

3. FCFD is compatible with many advanced knowledge distillation techniques with orthogonal contributions. When combined with these methods, FCFD can achieve even better performance.

## 2 METHOD

In this section, we start with a brief review of existing knowledge distillation methods in Sec. 2.1 for easier understanding of FCFD. In Sec. 2.2, we specify the proposed modules that align teacher and student features w.r.t. function. Finally, we introduce the complete training process in Sec. 2.3.

### 2.1 PRELIMINARIES

**Original KD:** Hinton et al. (2015) first clarified the concept of knowledge distillation (KD). In their work, probability distribution softened by a temperature parameter $\tau$ is used to transfer the knowledge from teacher to student:

$$\boldsymbol{p}(x) = \sigma\left(\boldsymbol{z}(\boldsymbol{x})/\tau\right), \tag{1}$$

where $x$ is input image, $\tau$ represents temperature used to soften the output distributions, $\sigma$ denotes the softmax function, and $\boldsymbol{z}$ is logit scores output by the penultimate layer of a neural network. KL-Divergence is then used to make the student learn from the teacher:

$$\mathcal{L}_{kd} = \tau^2 \mathbf{KL}\left(\boldsymbol{p}_t(x) \,\|\, \boldsymbol{p}_s(x)\right), \tag{2}$$

where $t$ and $s$ denote teacher and student, respectively.

**Feature Distillation:** Instead of using final probability distributions, feature distillation utilizes the knowledge in the intermediate features to guide the student. Compared with probability distribution, the intermediate feature has much larger volume, and theoretically it could provide more supervision. Considering a pair of teacher and student models, each of which is divided into $N$ sequential modules (stages) and a final linear classifier. The forward process of the networks is as follows:

$$\boldsymbol{p}_t(x) = C_t \circ M_t^N \circ \cdots \circ M_t^2 \circ M_t^1(x); \ \ \boldsymbol{p}_s(x) = C_s \circ M_s^N \circ \cdots \circ M_s^2 \circ M_s^1(x). \tag{3}$$

$M_t^{(\cdot)}$ and $M_s^{(\cdot)}$ denote the modules in teacher and student networks, $C_t$ and $C_s$ are their respective classifiers. Now consider the intermediate features:

$$F_t^k = M_t^k \circ \cdots \circ M_t^2 \circ M_t^1(x); \ \ F_s^k = M_s^k \circ \cdots \circ M_s^2 \circ M_s^1(x). \tag{4}$$

With a pre-defined list of distillation positions $\mathbb{K} \subseteq \{1, 2, \cdots, N\}$, at each position $k \in \mathbb{K}$, the teacher feature $F_t^k$ (hint layer) and the student feature $F_s^k$ (guided layer) are matched for distillation. With a bridge module $B_{st}^k$, which is a simple combination of one convolution and one BatchNorm (Ioffe & Szegedy, 2015) layer, as student transformation to match the shape of $F_t^k$, the simplest form of feature distillation can be achieved with the following appearance loss:

$$\mathcal{L}_{app}^k = \mathbf{L2}(F_t^k, B_{st}^k(F_s^k)). \tag{5}$$

## 2.2 FUNCTION-CONSISTENT FEATURE DISTILLATION

We propose FCFD, where both the numerical value of features and the information about later layers are combined together for faithful feature mimicking. An illustration of FCFD is shown in Fig. 2. Due to limited space, here we focus on the methodology on image classification. However, FCFD is also applicable to object detection, and relevant introduction is deferred to Appendix D.

As analyzed before, the appearance perspective (represented by $\mathcal{L}_{app}$) only is not satisfactory for feature matching. FCFD's solution to this problem is to investigate the function perspective through the lens of the network's later part: we feed both the teacher and the student features to the later part of the teacher/student network, and explicitly calculate the differences between consequent results as the function distance between the features. As the later part of the network defines the semantics of intermediate features, features making similar results after later network processing express similar semantics and are by definition functionally consistent. Depending on whether using the later part of the teacher or the student network as the lens, the proposed function-consistency-matching mechanism could be divided into two parts:

**Using teacher's later part as the lens:** given a matching position $k$, we hope that the transformed student feature $B_{st}^k(F_s^k)$ not only looks like $F_t^k$ (which is guaranteed by Eq. 5), but also plays similar function to $F_t^k$. Therefore, we feed both $B_{st}^k(F_s^k)$ and $F_t^k$ to the later part of the teacher network, and use the similarity in results to reflect feature's similarity in function:

$$\mathcal{L}_{func}^k = \sum_{l=k+1}^{N+1} \mathbf{d}_{fd}\bigg( M_t^l \circ \cdots \circ M_t^{k+1}\left(F_t^k\right), M_t^l \circ \cdots \circ M_t^{k+1}\left(B_{st}^k\left(F_s^k\right)\right) \bigg). \tag{6}$$

$M_t^{N+1}$ refers to $C_t$ and $M_s^{N+1}$ refers to $C_s$. $\mathbf{d}_{fd}$ is feature distance metric, which is set to KL-Divergence when applied to probability distributions ($l = N + 1$), and set to L2 distance otherwise. If teacher and student features are really semantically consistent, we would expect that after feeding them both to the teacher's later part, the output will be similar. Though L2 distance is used here, it does not match $F_t^k$ and $B_{st}^k(F_s^k)$ directly, but match them after processed by several later teacher modules. As a result, while $M_t^l \circ \cdots \circ M_t^{k+1}(F_t^k)$ and $M_t^l \circ \cdots \circ M_t^{k+1}(B_{st}^k(F_s^k))$ are matched w.r.t. appearance in Eq. 6, their underlying effect is to match $F_t^k$ and $B_{st}^k(F_s^k)$ w.r.t. function.

**Using student's later part as the lens:** Now that we have used the later part of the teacher model as the lens, it is natural to consider if the later part of student model can play similar role. We define $B_{ts}^k$, which makes $B_{ts}^k(F_t^k)$ of equal size to $F_s^k$, and consider to pass $B_{ts}^k(F_t^k)$ through the student's later modules. However, unlike the case for $\mathcal{L}_{func}$ where the teacher features are well-trained and informative enough to guide the student, now the transformed teacher features pass through randomly initialized bridge and student modules, so directly matching student features towards them is not reasonable. We thus make a different choice: we pass both $F_s^k$ and $B_{ts}^k(F_t^k)$ through the student's later modules *completely* (akin to the $l = N + 1$ case in Eq. 6) and obtain the corresponding final outputs $\boldsymbol{p}_s$ and $\boldsymbol{p}_{ts}^k$. We then match both of them towards $\boldsymbol{p}_t$, which is achieved through $\mathcal{L}_{kd}$ for $\boldsymbol{p}_s$, and $\mathcal{L}_{func'}^k$ for $\boldsymbol{p}_{ts}^k$:

$$\mathcal{L}_{func'}^k = \mathbf{KL}\left( \underbrace{C_t \circ M_t^N \circ \cdots \circ M_t^{k+1}(F_t^k)}_{\boldsymbol{p}_t} || \underbrace{C_s \circ M_s^N \circ \cdots \circ M_s^{k+1} \circ B_{ts}^k(F_t^k)}_{\boldsymbol{p}_{ts}^k} \right). \tag{7}$$

In this way, we encourage $F_s^k$ and $B_{ts}^k(F_t^k)$ to be functionally consistent w.r.t. student's later part.

**Synergy between Appearance and Function Perspectives:** $L_{func}$ and $L_{func'}$ consider the function perspective for feature matching, which is complementary to the appearance perspective considered by $L_{app}$: assuming appearance difference is small and constant, minimizing function distance

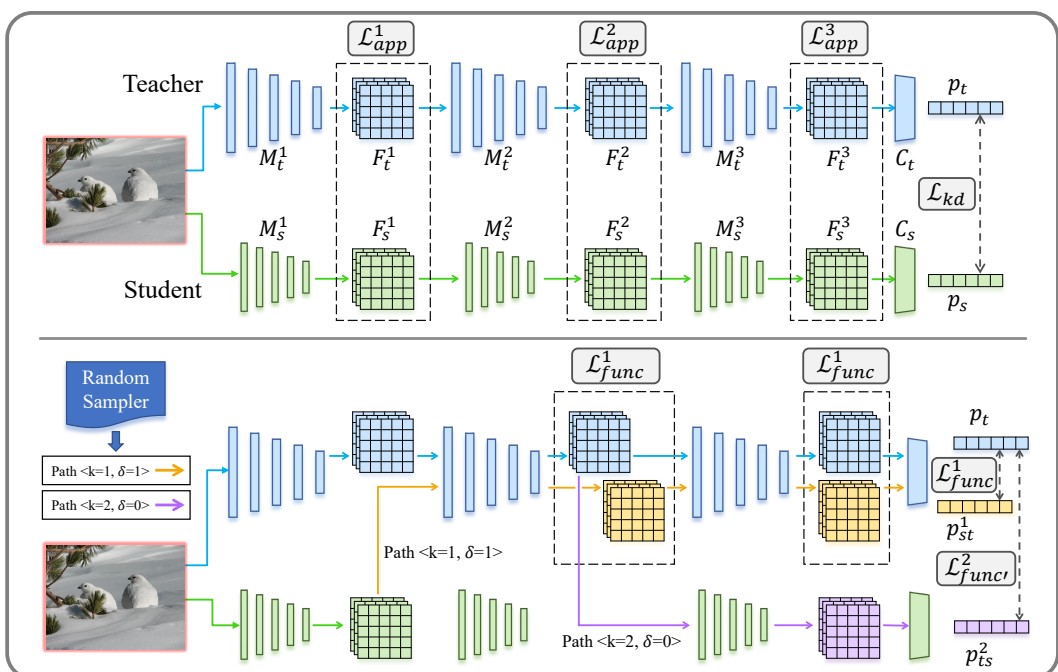

Figure 2: An overview of FCFD. Top: illustration of the traditional KD loss ($\mathcal{L}_{kd}$) and appearance-based feature matching loss ($\mathcal{L}_{app}$). Bottom: illustration of our proposed function matching losses $\mathcal{L}_{func}$ and $\mathcal{L}_{func'}$. Note that the three $\mathcal{L}_{func}^1$ terms in the figure sum up to complete $\mathcal{L}_{func}^1$. In each iteration, we randomly sample two paths (shown with yellow and purple arrows). Together with the pure teacher path (blue arrow) and pure student path (green arrow), data flow through totally four paths, and the losses are appended whenever applicable. Bridge modules are omitted for simplicity.

makes the student to pay more attention to sensitive directions for more task-related information. The mimicking then becomes more faithful, and the student becomes more likely to interpret rather than blindly memorize. On the other hand, as the mapping from intermediate features to final output could be many-to-one, minimizing appearance distance encourages the student features to be located in the same local area as the teacher features. In this way, more information can be exploited as guidance to the student. Ablation studies in Sec. 3.3 validates the aforementioned synergy.

## 2.3 FULL PIPELINE

Here we introduce the full pipeline of FCFD. See Fig. 2 for illustration. Before training, a list of candidate matching positions $\mathbb{K}$ is defined. According to Sec. 2.2, the complete loss function is:

$$\mathcal{L} = \mathcal{L}_{kd} + \mathcal{L}_{task} + \sum_{k \in \mathbb{K}} \left( \mathcal{L}_{app}^k + \mathcal{L}_{func}^k + \mathcal{L}_{func'}^k \right). \qquad (8)$$

However, directly matching all these features causes great resource cost. Therefore, in practice, a random sampling strategy is adopted for efficiency. Specifically, for any matching position $k$, each of $\mathcal{L}_{func}^k$ and $\mathcal{L}_{func'}^k$ involves an extra forward path. Suppose there are $K$ elements in $\mathbb{K}$, there are thus $2K$ total paths. We use pair $< k, \delta >$ to denote a specific path, where $\delta = 1$ if the path starts from the student and turns to the teacher for $\mathcal{L}_{func}^k$, and $\delta = 0$ when the path starts from the teacher and turns to the student for $\mathcal{L}_{func'}^k$. In each iteration, we sample a list of paths $\mathbb{S}$. Only these chosen paths will be propagated through, and the relevant losses will be computed:

$$\mathcal{L} = \mathcal{L}_{kd} + \mathcal{L}_{task} + \sum_{k \in \mathbb{K}} \mathcal{L}_{app}^k + \sum_{<k,\delta> \in \mathbb{S}} \left( \delta \mathcal{L}_{func}^k + (1 - \delta) \mathcal{L}_{func'}^k \right). \qquad (9)$$

Each loss term is scaled by corresponding weight hyper-parameter (which is omitted in Eq. 9) before summing up. In our default implementation, there are always 2 paths in $\mathbb{S}$. Combining with the pure teacher and student paths, in each iteration there are totally 4 paths involved. Note that some paths would share common data flows, and in such cases we only need to calculate them once.

Note that some BatchNorm layers may receive inputs calculated through different paths. For example, the BatchNorm layers in $M_t^3$ may receive inputs processed by $M_t^2 \circ M_t^1$, $M_t^2 \circ B_{st}^1 \circ M_s^1$, or $B_{st}^2 \circ M_s^2 \circ M_s^1$, respectively. We empirically find that since they have different distributions, using the default BatchNorm behavior to accumulate their running statistics together spoils the inference process. Therefore, for BatchNorm layers, we accumulate running statistics separately for input processed by different sets of modules. Meanwhile, the affine parameters are shared among all paths (Fig. 3).

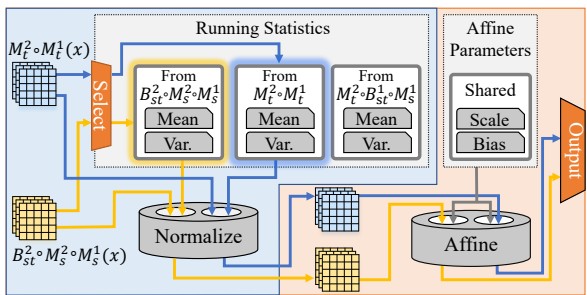

Figure 3: An illustration of the operations within a BatchNorm layer in $M_t^3$. Running Statistics are accumulated separately w.r.t. different input paths, while affine parameters are shared among all paths.

## 3 EXPERIMENT

In this section, we empirically validate the effectiveness of FCFD. In Sec. 3.1, we compare FCFD with state-of-the-art methods on image classification and object detection. In Sec. 3.2, we combine FCFD with advanced KD methods with orthogonal contributions to ours, and we show that in this way, FCFD can achieve further improvement. Ablative experiments are conducted in Sec. 3.3 to dissect the effect of each role in our method. Finally, in Sec. 3.4, we experimentally demonstrate that FCFD does improve teacher-student feature similarity from the function perspective. Details about our implementation and experiment settings are provided in Appendix.

### 3.1 COMPARATIVE EXPERIMENTS

**Image Classification on CIFAR100:** We first compare FCFD with six existing methods on CIFAR100 (Krizhevsky et al., 2009). For methods distilling last layer's output, we select KD (Hinton et al., 2015), CRD (Tian et al., 2020), DKD (Zhao et al., 2022) for comparison; for intermediate feature distillation, we compare with FitNet (Romero et al., 2015), VID (Ahn et al., 2019), and Review (Chen et al., 2021b). Following CRD (Tian et al., 2020), we choose CIFAR-style ResNet (He et al., 2016), Wide-ResNet (Zagoruyko & Komodakis, 2016), VGG (Simonyan & Zisserman, 2014), MobileNetV2 (Sandler et al., 2018), ShuffleNetV1 (Zhang et al., 2018a), and ShuffleNetV2 (Ma et al., 2018) as model architecture. Experiments are divided into two parts: in the first part, teacher and student have architectures of the same style; in the second part, they have different architectures.

Results are shown in Tab. 1. *For teacher and student with similar architecture*, FCFD averagely raises the student's performance by 3.70% over non-distillation baseline, and performs significantly better than vanilla KD, with an average improvement by 2.05%. Furthermore, FCFD outperforms state-of-the-art methods on 4 out of 5 teacher-student pairs, demonstrating the effectiveness of the proposed method with steady improvement. *For teacher and student with different architecture*, the improvement is more obvious. Comparing with non-distillation baseline, the student shows improvement by 6.78% on average. FCFD also averagely outperforms original KD by 3.57%. Moreover, FCFD is the best-performing method with all teacher-student pairs. Comparing with the second best method, Review, FCFD enjoys an average improvement by 0.66%, with the largest improvement by 1.11% (ResNet50-MobileNetV2).

**Image Classification on ImageNet:** To validate the efficacy of our method on large-scale datasets, we also compare FCFD with other methods on ImageNet (Deng et al., 2009). We compare with AT (Zagoruyko & Komodakis, 2017), OFD (Heo et al., 2019), SRRL (Yang et al., 2021), and four methods already considered in CIFAR100 experiments. ResNet (He et al., 2016) and MobileNet (Howard et al., 2017) are used as model architecture. The results are shown in Tab. 2. With the ResNet34-ResNet18 pair, comparing with non-distillation baseline and KD, FCFD achieves improvement by 2.5% and 1.6% top-1 accuracy, respectively. Furthermore, FCFD outperforms the second best method by 0.55% (v.s. DKD) over top-1 accuracy and by 0.2% (v.s. Review) over top-5 accuracy. With the ResNet50-MobileNet pair, the advantage of FCFD is even larger, with top-1 accuracy reaching 73.26%, which is 0.7% above the best existing method (Review). The results show

Table 1: Top-1 accuracy (%) on CIFAR100. **Bold** and underline denote the best and the second best results, respectively. Results of DKD and Review are quoted from their paper, and others are quoted from the CRD paper. * denotes the method includes $\mathcal{L}_{kd}$. FCFD results are averaged over three runs.

| Teacher and Student of Similar Architectures | | | | |
|---|---|---|---|---|
| Teacher | WRN-40-2 | WRN-40-2 | ResNet56 | ResNet32x4 | VGG13 |
| Student | WRN-16-2 | WRN-40-1 | ResNet20 | ResNet8x4 | VGG8 |
| Teacher | 75.61 | 75.61 | 72.34 | 79.42 | 74.64 |
| Student | 73.26 | 71.98 | 69.06 | 72.50 | 70.36 |
| KD* | 74.92 | 73.54 | 70.66 | 73.33 | 72.98 |
| CRD* | 75.64 | 74.38 | 71.63 | 75.46 | 74.29 |
| DKD | 76.24 | 74.81 | **71.97** | 76.32 | 74.68 |
| FitNet | 73.58 | 72.24 | 69.21 | 73.50 | 71.02 |
| VID | 74.11 | 73.30 | 70.38 | 73.09 | 71.23 |
| Review | 76.12 | 75.09 | 71.89 | 75.63 | 74.84 |
| Ours (w/o $\mathcal{L}_{kd}$) | **76.34** | **75.43** | 71.68 | **76.80** | **74.86** |
| Ours* | 76.43 | 75.46 | 71.96 | 76.62 | 75.22 |

| Teacher and Student of Different Architectures | | | | |
|---|---|---|---|---|
| Teacher | ResNet32x4 | WRN-40-2 | VGG13 | ResNet50 | ResNet32x4 |
| Student | ShuffleV1 | ShuffleV1 | MobileNetV2 | MobileNetV2 | ShuffleV2 |
| Teacher | 79.42 | 75.61 | 74.64 | 79.34 | 79.42 |
| Student | 70.50 | 70.50 | 64.60 | 64.60 | 71.82 |
| KD* | 74.07 | 74.83 | 67.37 | 67.35 | 74.45 |
| CRD* | 75.11 | 76.05 | 69.73 | 69.11 | 75.65 |
| DKD | 76.45 | 76.70 | 69.71 | 70.35 | 77.07 |
| FitNet | 73.59 | 73.73 | 64.14 | 63.16 | 73.54 |
| VID | 73.38 | 73.61 | 65.56 | 67.57 | 73.40 |
| Review | 77.45 | 77.14 | 70.37 | 69.89 | 77.78 |
| Ours (w/o $\mathcal{L}_{kd}$) | **78.12** | **77.81** | **70.67** | **71.07** | **78.20** |
| Ours* | 78.12 | 77.99 | 70.65 | 71.00 | 78.18 |

Table 2: Accuracy (%) on ImageNet validation set. Setting (a): ResNet-34 as teacher and ResNet18 as student. Setting (b): ResNet-50 as teacher and MobileNet as student. **Bold** and underline denote the best and the second best results, respectively. * denotes the method includes $\mathcal{L}_{kd}$. For both settings, the first and the second row show the top-1 and the top-5 accuracy, respectively.

| setting | Teacher | Student | KD* | AT | OFD | CRD* | Review | DKD | SRRL | Ours | Ours* |
|---|---|---|---|---|---|---|---|---|---|---|---|
| (a) | 73.31 | 69.75 | 70.66 | 70.69 | 70.81 | 71.17 | 71.61 | 71.70 | 71.73 | **72.24** | 72.25 |
| | 91.42 | 89.07 | 89.88 | 90.01 | 89.98 | 90.13 | 90.51 | 90.41 | 90.60 | **90.74** | 90.71 |
| (b) | 76.16 | 68.87 | 68.58 | 69.56 | 71.25 | 71.37 | 72.56 | 72.05 | 72.49 | **73.37** | 73.26 |
| | 92.86 | 88.76 | 88.98 | 89.33 | 90.34 | 90.41 | 91.00 | 91.05 | 90.92 | **91.35** | 91.24 |

that the proposed distillation mechanism could steadily produce high-quality models regardless of dataset volume and task complexity.

**Object Detection on MS-COCO:** We further validate the effectiveness of FCFD on the MS-COCO (Lin et al., 2014) object detection task, using Faster-RCNN-FPN with different backbone models. FGFI (Wang et al., 2019), ICD (Kang et al., 2021), and Review (Chen et al., 2021b) are adopted for comparison. Results are shown in Tab. 3. FCFD enjoys an average improvement over non-distillation baseline by 4.03% mAP, and consistently outperforms existing methods.

Considering both image classification and object detection, we find that FCFD has greater advantage when teacher and student models are of different architectures, which may indicate that function consistency between teacher and student features are harder to be guaranteed without explicit optimization when the two models have different inductive bias.

### 3.2 COMPATIBILITY WITH EXISTING METHODS

In this section, we combine the proposed FCFD with existing knowledge distillation techniques with orthogonal contributions. Two recent works are considered: DKD (Zhao et al., 2022) and

Table 3: Compare FCFD with other knowledge distillation methods on object detection. Models are trained on MS-COCO `train2017` and tested on MS-COCO `val2017`.

| | Method | mAP | AP50 | AP75 | APl | APm | Aps |
|---|---|---|---|---|---|---|---|
| Teacher | Faster R-CNN w/ R101-FPN | 42.04 | 62.48 | 45.88 | 54.60 | 45.55 | 25.22 |
| Student | Faster R-CNN w/ R18-FPN | 33.26 | 53.61 | 35.26 | 43.16 | 35.68 | 18.96 |
| | w/ FGFI | 35.44 (+2.18) | 55.51 | 38.17 | 47.34 | 38.29 | 19.04 |
| | w/ ICD | 35.90 (+2.64) | 56.02 | 38.75 | 46.83 | 38.78 | 20.22 |
| | w/ Review | 36.75 (+3.49) | 56.72 | 34.00 | 49.58 | 39.51 | 19.42 |
| | **w/ Our Method** | **37.37 (+4.11)** | **57.60** | **40.34** | **50.33** | **40.23** | **19.84** |
| Teacher | Faster R-CNN w/ R101-FPN | 42.04 | 62.48 | 45.88 | 54.60 | 45.55 | 25.22 |
| Student | Faster R-CNN w/ R50-FPN | 37.93 | 58.84 | 41.05 | 49.10 | 41.14 | 22.44 |
| | w/ FGFI | 39.44 (+1.51) | 60.27 | 43.04 | 51.97 | 42.51 | 22.89 |
| | w/ ICD | 40.39 (+2.46) | 61.14 | 43.97 | 52.22 | 44.16 | 23.69 |
| | w/ Review | 40.36 (+2.43) | 60.97 | 44.08 | 52.87 | 43.81 | 23.60 |
| | **w/ Our Method** | **40.42 (+2.49)** | **61.01** | **43.66** | **53.77** | **43.61** | **24.63** |
| Teacher | Faster R-CNN w/ R50-FPN | 40.22 | 61.02 | 43.81 | 51.98 | 43.53 | 24.16 |
| Student | Faster R-CNN w/ MV2-FPN | 29.47 | 48.87 | 30.90 | 38.86 | 30.77 | 16.33 |
| | w/ FGFI | 31.16 (+1.69) | 50.68 | 32.92 | 42.12 | 32.63 | 16.73 |
| | w/ ICD | 32.88 (+3.41) | 52.56 | 34.93 | 42.65 | 34.73 | 18.19 |
| | w/ Review | 33.71 (+4.24) | 53.15 | 36.13 | 46.47 | 35.81 | 16.77 |
| | **w/ Our Method** | **34.97 (+5.50)** | **55.04** | **37.51** | **47.60** | **37.09** | **18.82** |

Table 4: Combine FCFD with DKD (Zhao et al., 2022) and SimKD (Chen et al., 2022). Top-1 accuracy (%) on CIFAR-100 is provided. Result format of the DKD and SimKD: result reported by original authors (our reimplemented result). - means not reported.

| Teacher | WRN-40-2 | WRN-40-2 | ResNet32x4 | ResNet32x4 | ResNet32x4 |
|---|---|---|---|---|---|
| Student | WRN-16-2 | WRN-40-1 | ResNet8x4 | ShuffleV1 | ShuffleV2 |
| Teacher | 75.61 | 75.61 | 79.42 | 79.42 | 79.42 |
| Student | 73.26 | 71.98 | 72.50 | 70.50 | 71.82 |
| FCFD | **76.43** | 75.46 | 76.62 | 78.12 | 78.18 |
| DKD | 76.24 (76.16) | 74.81 (74.99) | 76.32 (76.23) | 76.45 (76.60) | 77.07 (76.88) |
| FCFD+DKD | 76.37 | 75.4 | 76.95 | 78.20 | 78.79 |
| SimKD | - (75.73) | 75.56 (75.32) | 78.08 (77.98) | 77.18 (77.41) | 78.39 (77.74) |
| FCFD+SimKD | 76.32 | **76.08** | **78.82** | **78.29** | **79.33** |

SimKD (Chen et al., 2022). We show that when combined with these methods, FCFD could achieve even better performance than its original version, demonstrating the great potential of FCFD. All experiments are conducted on CIFAR100. The introduction of DKD and SimKD, as well as the technical details on how to combine these two methods with FCFD are provided in Appendix C.

Tab. 4 shows the results. *When combined with DKD*, which can be considered as an enhanced version of traditional KL-Divergence loss function, FCFD can at most obtain an improvement by 0.61% (ResNet32x4-ShuffleV2). One significant advantage of DKD loss is that it induces no additional cost during both training and inference. Therefore, this combination is very friendly to practical application. *When combined with SimKD*, the improvement is greater, with the magnitude at most 2.2% (ResNet32x4-ResNet8x4). Note that for SimKD, the last bridge module is still required during inference, and thus it involves extra inference cost. However, considering the introduced performance improvement, it is still worthwhile in many cases.

The improvement is much larger when the teacher is ResNet32x4 rather than WRN-40-2, one possible explanation is that the performance of WRN-40-2 itself is relatively low and has barricaded the progress of the student. As shown in Tab. 4, with either WRN-16-2 or WRN-40-1 as student, the performance of the student has approached or even surpassed the teacher.

### 3.3 ABLATION STUDY

In this section, we conduct ablative experiments to analyze the role played by each component adopted in FCFD. Experiment results are shown in Tab. 5.

Table 5: Ablative experiments to investigate the effect of each component in FCFD. Experiments are conducted on CIFAR-100, with ResNet32x4 as teacher. Top-1 accuracy (%) is provided.

| # | Ablation | $\mathcal{L}_{kd} + \mathcal{L}_{task}$ | $\mathcal{L}_{app}$ | $\mathcal{L}_{func}$ | $\mathcal{L}_{func'}$ | ResNet8x4 | ShuffleV1 | ShuffleV2 |
|---|---|---|---|---|---|---|---|---|
| ① | Baseline | ✓ | | | | 74.77 | 74.19 | 75.42 |
| ② | Appearance only | ✓ | ✓ | | | 75.89 | 76.77 | 76.49 |
| ③ | Function only | ✓ | | ✓ | ✓ | 75.93 | 76.73 | 77.35 |
| ④ | w/o $\mathcal{L}_{func'}$ | ✓ | ✓ | ✓ | | 76.32 | 77.92 | 78.02 |
| ⑤ | w/o $\mathcal{L}_{func}$ | ✓ | ✓ | | ✓ | 76.62 | 77.19 | 77.43 |
| ⑥ | FCFD | ✓ | ✓ | ✓ | ✓ | **76.62** | **78.12** | **78.18** |

Table 6: Further analysis to investigate student-teacher intermediate feature similarity in terms of function. Intermediate features extracted by teacher and student models are fed to exit branches, and the top-1 accuracy (%) of the outputs are reported. Dataset: CIFAR100. Teacher: ResNet32x4

| Position | After the second module | | | After the third module | | |
|---|---|---|---|---|---|---|
| Student | ResNet8x4 | ShuffleV1 | ShuffleV2 | ResNet8x4 | ShuffleV1 | ShuffleV2 |
| Teacher | 78.68 | 78.68 | 78.68 | 78.37 | 78.37 | 78.37 |
| $\mathcal{L}_{kd} + \mathcal{L}_{task} + \mathcal{L}_{app}$ | 78.00 | 77.21 | 77.69 | 76.14 | 73.61 | 73.80 |
| FCFD | 78.33 | 78.20 | 78.61 | 77.31 | 75.45 | 76.96 |

**Synergistic appearance and function perspectives:** Considering ①②③ in Tab. 5, the results show that matching intermediate representations from appearance and function perspectives are *both* effective approaches to improve the student's performance. After taking ⑥ into consideration, we can find that these two perspectives can be combined together for further improvement. The results supports our expectation that the two perspectives are synergistic in intermediate feature distillation.

**Complementary $\mathcal{L}_{func}$ and $\mathcal{L}_{func'}$:** considering ④⑤⑥ in Tab. 5, we can draw the conclusion that both $\mathcal{L}_{func}$ and $\mathcal{L}_{func'}$ are indispensable in FCFD. Specifically, in some cases (*e.g.*, ShuffleV1 and ShuffleV2), $\mathcal{L}_{func}$ brings in greater improvement, while the opposite is true in other cases (*e.g.*, ResNet8x4). Nevertheless, when both of them are adopted, the result is always better than using either only. Note that since we always sample 2 paths per iteration (*i.e.*, there are two elements in $\mathbb{S}$), considering both $\mathcal{L}_{func}$ and $\mathcal{L}_{func'}$ brings about *no extra training cost* except the negligible static storage load to store additional bridges.

## 3.4 FURTHER ANALYSIS

In this section, we empirically show that FCFD does help enhance teacher-student intermediate feature similarity in terms of semantics. Specifically, we append exit branches on top of intermediate positions of the pre-trained teacher model, and we train the branches with the original teacher model fixed. The training task is the simple hard label classification task. In this way, we create a mapping that extracts knowledge from teacher features and estimates the hard label. Note that we make the exit branches strong enough (even larger than the later part of teacher after the given position), so that the information implied in teacher features could be fully exploited. We then directly feed student features (after passing the bridge module) to these exit branches, and record the accuracy of outputs. The results are shown in Tab. 6. Without any exposure of student features during the training of exit branches, comparing with simple appearance-based features distillation, using FCFD raises the accuracy of the output. It means that by matching features w.r.t. function, the student better comprehends the semantics under the teacher's features, and shares more information that are task relevant. In this way, the student features become closer to teacher features w.r.t. the semantics they express. Details about the experiment settings are provided in Appendix A.2.

## 4 CONCLUSION

In this paper, we propose a novel knowledge distillation method called FCFD. FCFD aims to match the functional similarity between teacher and student features, in hopes that the student can benefit more from the distillation process. Extensive experiments demonstrate that FCFD steadily outperforms state-of-the-art methods, and can also be combined with some existing methods for further performance improvement. Furthermore, ablation studies are provided to dissect the inner workings.

## 5 ACKNOWLEDGEMENT

This work was partially supported by the National Key R&D Program of China (2021ZD0111901), the Natural Science Foundation of China (No. 62122074).

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

# A  EXPERIMENT DETAILS

## A.1  BASIC SETTINGS

Here we introduce the basic settings of our experiments. These settings are used for comparative experiments in Sec. 3.1, and are also used in other experiments unless otherwise specified.

**CIFAR100** The CIFAR100 (Krizhevsky et al., 2009) dataset consists of 60K images from 100 categories with size of $32 \times 32$. In the standard protocol, 50k images are used for training and 10k for testing. Following CRD (Tian et al., 2020), we train all the models for 240 epochs, and the learning rate is decayed by a factor of 10 at 150, 180, and 210 epochs, respectively. The initial learning rate is 0.01 for MobileNetV2, ShuffleNet, ShuffleNetV2, and is 0.05 for other student models. The batch size is 64, and SGD optimizer with a 0.0005 weight decay and 0.9 momentum is adopted. For all experiments, the weights for $\mathcal{L}_{task}$ and $\mathcal{L}_{kd}$ are set to 1. The KL-Divergence term in $\mathcal{L}_{func}$ and $\mathcal{L}_{func'}$ share the same weight that is either 1 or 0.2. $\mathcal{L}_{app}$ and $L2$ terms in $\mathcal{L}_{func}$ share the same weight, which is either 0.02 or 5.0. For $\mathcal{L}_{kd}$, the KL term in $\mathcal{L}_{func}$, and the KL term in $\mathcal{L}_{func'}$, the same temperature (Hinton et al., 2015) is used and is either 4 or 8.

**ImageNet** The ImageNet (Deng et al., 2009) dataset consists of 1.28 million training images and 50k validation images from 1000 categories. Following the mainstream settings, all methods are trained on the entire training set and evaluated on the single-crop validation set. The input image resolution is $224 \times 224$ for both training and evaluation. Following mainstream settings (Chen et al., 2021; Tian et al., 2020), we train all the models for 100 epochs. The initial learning rate is 0.1 and is decayed by a factor of 10 at 30, 60, and 90 epochs, respectively. We run experiments on one Tesla-V100 GPU with batch size 256 and initial learning rate 0.1. Automatic Mixed Precision (AMP) provided by PyTorch is used for acceleration. An SGD optimizer with 0.0001 weight decay and 0.9 momentum is adopted. The weights for $\mathcal{L}_{task}$, $\mathcal{L}_{kd}$, $\mathcal{L}_{func'}$, and the KL-Divergence term in $\mathcal{L}_{func}$, are set to 1, and the weights for $\mathcal{L}_{app}$ and the L2 term in $\mathcal{L}_{func}$ are set to 0.02 for ResNet18 and 5 for MobileNet. The temperature for $\mathcal{L}_{kd}$, the KL term in $\mathcal{L}_{func}$, and $\mathcal{L}_{func'}$, is set to 1.

The design of the bridge module $B(\cdot)$ follows CRD (Tian et al., 2020). Specifically, The bridge module is composed of exactly one convolution layer and one BatchNorm Layer. When the mimicking target is post-ReLU feature, and additional leaky ReLU activation is appended. The kernel size is 3x3, and is of stride 2 when the target feature is 2 times smaller, and is of stride 1 when the target is as large as the source feature. When the target is 2 times larger than the source, transposed convolution with kernel size 4 and stride 2 is used.

## A.2  SETTINGS FOR FURTHER ANALYSIS

Here we introduce the details of the experiments conducted in Sec. 3.4. We first select two intermediate positions. Specifically, considering that ResNet32x4, ResNet8x4, ShuffleV1, and ShuffleV2 are all composed of four cascaded modules, we select the end of the second and the third module. For all the experiments, we first take a ResNet32x4 model pre-trained on CIFAR100 (which is the teacher model for distillation), and add extra exit branches at the aforementioned positions. For each position, the structure of the exit branch is the same as the later part of a ResNet50x4 model after this position. More specifically, the structure of the exit branch at the end of the second module is the same as the combination of the last two modules of ResNet50x4, and the structure of the exit branch at the end of the third module is the same as the last module of ResNet50x4. In this way, we guarantee that the exit branch is strong enough and could well capture the task relevant information implied in teacher features.

The exit branches are then optimized, with teacher features as input, to estimate the hard label. Cross entropy loss is used as loss function. Similar to settings in Sec. A.1, the training process lasts for 240 epochs. The initial learning rate is 0.05 and is decayed by a factor of 10 at 150, 180, and 210 epochs, respectively. Batch size is 64. SGD optimizer with a 0.0005 weight decay and 0.9 momentum is adopted. Note that the ResNet32x4 backbone model is frozen in this process.

After training, we directly feed student features (transformed by the bridge modules) to these exit branches, and test the performance on CIFAR100 test set. Since the running statistics in the Batch-Norm layers in the exit branches are accumulated with teacher features as input, we re-calibrate the running statistics on CIFAR100 training set before testing.

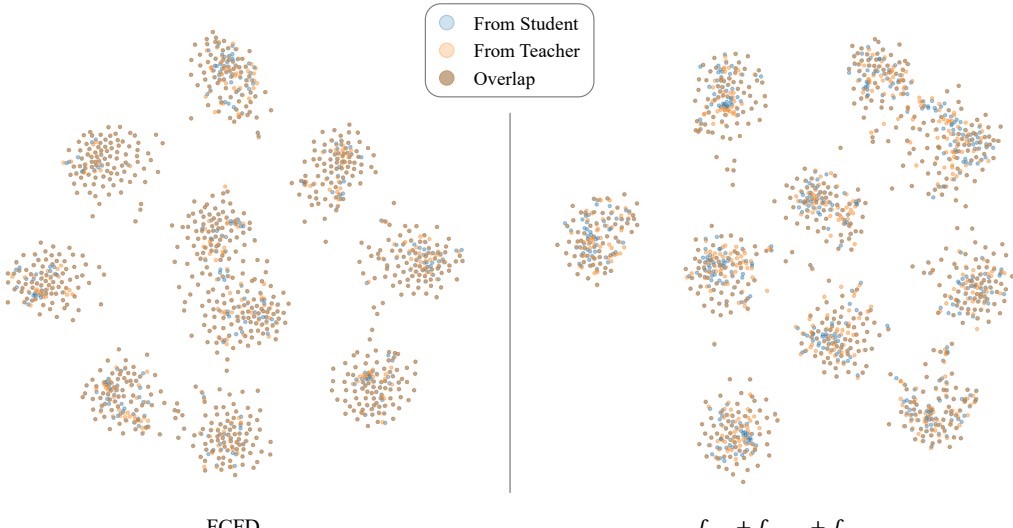

FCFD $\qquad$ $\mathcal{L}_{kd} + \mathcal{L}_{task} + \mathcal{L}_{app}$

Figure 4: Visualization of the semantic similarity between teacher and student intermediate features. We use images belonging to the first 10 classes in the CIFAR-100 test set. Teacher: ResNet32x4. Student: ShuffleV2. Student and Teacher features originally show light blue ◯ and light red ◯. When the student feature overlaps with the teacher feature, the overlapping region will become dark brown ●. With FCFD, fewer light blue and light red points can be observed, signifying that more student features are similar to their corresponding teacher features. **Best viewed in color with zoom in.**

## B VISUALIZATION FOR FEATURE SIMILARITY

We have empirically shown in Sec. 3.4 that FCFD improves the semantic similarity between teacher and student features. In this section, we provide a visualization to further demonstrate this effect. We feed the teacher and student intermediate features into the exit branches mentioned in Sec. 3.4 and Sec. A.2. We then collect the corresponding penultimate features and use t-SNE to project these features into points in $\mathbb{R}^2$. The results are shown in Fig. 4. With only appearance-based feature matching, there are a lot of light-blue (◯ From Student) and light-red (◯ From Teacher) points. These points show the light color because they do not overlap with the nearest points showing the other light color, *i.e.*, the student features lie relatively far from the nearest teacher features and vice versa. In contrast, with FCFD there are fewer light-color points and more dark-brown (● Overlap) points, which means more student features are similar to their corresponding teacher features. The results support our claim in Sec. 3.4.

## C DETAILS ON COMBINING FCFD WITH EXITING METHODS

In this section, we briefly introduce how we combine DKD (Zhao et al., 2022) and SimKD (Chen et al., 2022) with FCFD.

### C.1 COMBINE WITH DKD

The key idea of DKD (Zhao et al., 2022) is to decompose traditional knowledge distillation into two parts: Target Class Knowledge Distillation(TCKD), which matches the teacher's and student's binary probabilities of the target class, and Non-Target Class Knowledge Distillation(NCKD), which optimizes the similarity between the teacher's and student's probabilities among non-target classes. In traditional KD, the weights of TKCD and NCKD terms are coupled with teacher's confidence on the target class. In contrast, DKD propose to assign constant weights to these terms.

The DKD loss can be considered as an enhanced version of KL-Divergence. Therefore, to combine DKD with FCFD, we can simply replace with DKD loss the KL-Divergence loss in $\mathcal{L}_{kd}$, $\mathcal{L}_{func}$, and $\mathcal{L}_{func'}$. In our experiments, we set weight for TCKD to 1, and set weight for NCKD to 2. Other training settings are aligned with those in Sec. A.1.

## C.2   COMBINE WITH SIMKD

The key idea of SimKD is to make the student reuse the teacher's classifier layer. In this way, the forward process of the student is:

$$\boldsymbol{p}_{s(\text{simkd})}(x) = C_t \circ B_{st}^N \circ M_s^N \circ \cdots \circ M_s^2 \circ M_s^1(x) \tag{10}$$

Note that unlike other methods, for SimKD, the last bridge module, $B_{st}^N$, is still required during inference, and thus it involves extra inference cost.

To combine SimKD with FCFD, whenever we originally feed feature to $C_s$, now as the substitute, we feed feature to $C_t \circ B_{st}^N$. Specifically, the computation of $\boldsymbol{p}_s(x)$ and $\boldsymbol{p}_{ts}^k(x)$ now ends up with $C_t \circ B_{st}^N$ rather than $C_s$. All other settings, including target function, training strategy, are not changed. During inference, the forward path of the student is the same as Eq. 10.

As suggested in Chen et al. (2022), the capacity of the bridge module $B_{st}^N$ has significant impact on the final accuracy, and there exists an accuracy-latency trade-off. To make experiment results more informative, when combined with SimKD, we change $B_{st}^N$ to exactly match the architecture used in the original SimKD paper, and the squeeze factor is set to 2. Note that bridges modules other than $B_{st}^N$ are not changed for simplicity.

## D   DETAILS ABOUT OBJECT DETECTION

In Sec. D.1, we introduce the methodology of migrating FCFD to object detection. In Sec. D.2, we show the implementation details.

### D.1   METHOD

On object detection, we do *not* match features by feed the transformed student (teacher) backbone features to the later part of the teacher (student) backbone model. The reason are twofold: 1) whereas the propagation of image classification models is unidirectional, intermediate features calculated by detector's backbone model may be revisited long after its first usage due to the existence of FPN (Lin et al., 2017), region proposal networks (RPN), and ROI heads. Therefore, directly matching intermediate features calculated by detectors' backbone models w.r.t. function will make the subsequent propagation process complicated, and could involve great extra overhead; 2) In existing works, the common practice for conducting feature distillation on detectors is to match the features output by FPN (Chen et al., 2021; Wang et al., 2019; Yang et al., 2022), rather than by the backbone. When considering how to functionally align FPN features, the situation becomes clearer and easier. Therefore, the detection version of FCFD is designed to match FPN features.

Denote the list of teacher features output by FPN as $\widetilde{\mathbb{F}}_t = \{\widetilde{F}_t^k\}_{k=1}^K$, and the list of student features output by FPN as $\widetilde{\mathbb{F}}_s = \{\widetilde{F}_s^k\}_{k=1}^K$. Here with slight abuse of notation, we use $K$ to denote the number of FPN features. Given an input image $X$ and a region proposal $r$, the ROI head (denoted as ROIH) selects the corresponding part of feature from $\widetilde{\mathbb{F}}$, and outputs the probability $p(x,r)$ and positional offset $\delta(x,r)$ corresponding to all candidate classes:

$$< p_t(x,r), \delta_t(x,r) > = \text{ROIH}_t(r, \widetilde{\mathbb{F}}_t) \tag{11}$$

$$< p_s(x,r), \delta_s(x,r) > = \text{ROIH}_s(r, \widetilde{\mathbb{F}}_s) \tag{12}$$

subscript $t$ and $s$ denote teacher and student, respectively. Eq. 11 and Eq. 12 show that the function of $\widetilde{\mathbb{F}}$ is to serve as the input to the ROI head module. Following the same idea as our method on classification, we match $\widetilde{\mathbb{F}}_s$ and $\widetilde{\mathbb{F}}_t$ w.r.t. both appearance and function simultaneously. Appearance matching is achieved by:

$$\mathcal{L}_{app}^k = \textbf{L2}(\widetilde{F}_t^k, B_{st}^k(\widetilde{F}_s^k)) \tag{13}$$

Where $B_{st}$ is the student-to-teacher bridge module. For function matching, we first use the teacher's ROI head (ROIH$_t$) as the lens to measure the functional similarity between teacher and student features: we feed transformed student features to ROIH$_t$:

$$< p_{st}(x,r), \delta_{st}(x,r) > = \text{ROIH}_t(r, \{B_{st}^k(\widetilde{F}_s^k)|\widetilde{F}_s^k \in \widetilde{\mathbb{F}}_s\}) \tag{14}$$

and then align the results with those given by the teacher:

$$\mathcal{L}_{func} = \sum_{r \in \mathbb{R}} (\mathbf{KL}(p_t(x,r)||p_{st}(x,r)) + \mathbf{L2}(\delta_t(x,r), \delta_{st}(x,r))) \tag{15}$$

$\mathbb{R}$ denotes the set of region proposals fed to ROI heads. We then use the ROI head of the student ($\texttt{ROIH}_s$) as the lens for function matching: we feed both student features and transformed teacher features to $\texttt{ROIH}_s$, and optimize both results towards that given by the teacher. For the student features, the optimization is achieved by $\mathcal{L}_{kd}$:

$$\mathcal{L}_{kd} = \sum_{r \in \mathbb{R}} (\mathbf{KL}(p_t(x,r)||p_s(x,r)) + \mathbf{L2}(\delta_t(x,r), \delta_s(x,r))) \tag{16}$$

and for the transformed teacher features, the optimization is achieved by $\mathcal{L}_{func'}$

$$< p_{ts}(x,r), \delta_{ts}(x,r) > = \texttt{ROIH}_s(r, \{B_{ts}^k(\widetilde{F}_t^k)|\widetilde{F}_t^k \in \widetilde{\mathbb{F}}_t\}) \tag{17}$$

$$\mathcal{L}_{func'} = \sum_{r \in \mathbb{R}} (\mathbf{KL}(p_t(x,r)||p_{ts}(x,r)) + \mathbf{L2}(\delta_t(x,r), \delta_{ts}(x,r))) \tag{18}$$

$B_{ts}$ denotes the teacher-to-student bridge module.

In conclusion, the complete loss function for our detection-version FCFD is:

$$\mathcal{L} = \mathcal{L}_{task} + \sum_{k=1}^{K} \mathcal{L}_{app}^k + \mathcal{L}_{func} + \mathcal{L}_{func'} + \mathcal{L}_{kd} \tag{19}$$

$\mathcal{L}_{task}$ here denotes the original loss terms used in object detection. Since the ROI heads generally contain few parameters and the propagation is fast and efficient, the induced extra overhead for directly optimizing Eq. 19 is limited and acceptable. Therefore, no random sampling mechanism like that proposed in Sec. 2.3 is applied for object detection.

### D.2 IMPLEMENTATION DETAILS

Our implementation is based on Detectron2 (Wu et al., 2019). The weights for all of our proposed loss terms are set to 1. Temperature is set to 1 when calculating the KL-Divergence between two distributions. Automatic Mixed Precision (AMP) provided by PyTorch is used for acceleration. Batch size is set to 8 and initial learning rate is set to 0.01. For fair comparison with existing methods, we adopt the $\texttt{1x}$ training procedure. All experiments are conducted on one Tesla-V100 GPU. Other settings are inherited from Detectron2 and left unchanged.

## E MORE EXPERIMENTS ON OBJECT DETECTION

Inheriting strategy, which initializes the student's non-backbone modules with the teacher's pre-trained parameters, was proposed in ICD (Kang et al., 2021) and was shown to boost the student's performance clearly. We show in Tab. 7 that FCFD is also compatible with this strategy. Note that the inheriting strategy is applicable only when the output feature from the teacher's and the student's backbones have the same shape, so we report the results of the ResNet101-ResNet50 pair.

Following Review (Chen et al., 2021) and ICD (Kang et al., 2021), for object detection, we conduct experiments on Detectron2 (Wu et al., 2019). Therefore, in Tab. 3 we compare with methods implemented on the same framework for fairness. Nevertheless, we also notice that some other works (Yang et al., 2022; Dai et al., 2021) report their performance on MMDetection (Chen et al., 2019). Here we re-implement our FCFD on MMDetection and compare it with these works to provide readers with a comprehensive view of FCFD's performance, see Tab. 8 for results.

## F ONLINE KD

We also validate the proposed method in the context of online knowledge distillation (Zhang et al., 2018). Specifically, both the student and the teacher are randomly initialized before training. During

Table 7: Results of distillation with inheriting strategy. † denotes the inheriting strategy.

| | Method | mAP | AP50 | AP75 | APl | APm | Aps |
|---|---|---|---|---|---|---|---|
| Teacher | Faster R-CNN w/ R101-FPN | 42.04 | 62.48 | 45.88 | 54.60 | 45.55 | 25.22 |
| Student | Faster R-CNN w/ R50-FPN | 37.93 | 58.84 | 41.05 | 49.10 | 41.14 | 22.44 |
| | **w/ Our Method** | **40.42 (+2.49)** | **61.01** | **43.66** | **53.77** | **43.61** | **24.63** |
| | **w/ Our Method†** | **40.96 (+3.03)** | **61.78** | **44.79** | **54.55** | **44.40** | **24.19** |

Table 8: Comparison with existing methods on MMDetection. Results other than ours are quoted from Yang et al. (2022). † denotes the inheriting strategy.

| | Method | mAP | APl | APm | Aps |
|---|---|---|---|---|---|
| Teacher | Faster R-CNN w/ R101-FPN | 39.8 | 52.8 | 43.6 | 22.5 |
| Student | Faster R-CNN w/ R50-FPN | 38.4 | 50.3 | 42.1 | 21.5 |
| | w/ FGFI (Wang et al., 2019) | 39.3 | 52.2 | 42.3 | 22.5 |
| | w/ GID (Dai et al., 2021) | 40.2 | 53.2 | 44.0 | 22.7 |
| | w/ FGD (Yang et al., 2022) | 40.4 | 53.5 | 44.5 | 22.8 |
| | **w/ Our Method** | **40.8 (+2.4)** | **53.4** | **44.9** | **23.0** |
| | w/ FGD† | 40.5 | 53.2 | 44.7 | 22.6 |
| | **w/ Our Method†** | **40.9 (+2.5)** | **53.9** | **44.8** | **23.4** |

training, the teacher and the student learn mutually from each other. Apart from the loss function in Eq. 9, the task loss w.r.t. teacher output and hard label, as well as KD loss that the teacher learns from the student, are appended.

The training settings are the same as those in Sec. A.1. DML (Zhang et al., 2018) and CRD (Tian et al., 2020) are used for comparison. For DML, we re-implement the method for results. For CRD, we directly quote the results provided in the appendix of their paper. As shown in Tab. 9, FCFD not only performs well on offline knowledge distillation, but also outperforms existing methods on online knowledge distillation. This shows that the idea of matching intermediate features w.r.t. function is effective in a wide range of scenarios.

# G    ABLATION OVER THE DESIGN OF $\mathcal{L}_{func}$ AND $\mathcal{L}_{func'}$

In this section, we delve deep into the design choices of $\mathcal{L}_{func}$ and $\mathcal{L}_{func'}$.

Considering Eq. 6, $\mathcal{L}_{func}$ can be divided into two parts: the KL-Divergence loss defined on the final outputs ($l = N + 1$), and L2 losses defined on earlier layers ($l < N + 1$). We denoted the aforementioned KL-Divergence part as $\mathcal{L}_{func-KL}$, and the L2 part as $\mathcal{L}_{func-L2}$.

Considering Eq. 7, for reasons stated in Sec. 2.2, it only involves a KL term. To investigate if feature matching loss similar to $\mathcal{L}_{func-L2}$ is useful in the case of $\mathcal{L}_{func'}$, we define the following loss:

$$\mathcal{L}_{func'-L2}^k = \sum_{l=k+1}^{N} \mathbf{L2}\left( M_s^l \circ \cdots \circ M_s^{k+1}\left( B_{ts}^k\left( F_t^k \right)\right), M_s^l \circ \cdots \circ M_s^{k+1}\left( F_s^k \right) \right). \quad (20)$$

We conduct experiments to analyze these terms. Results are shown in Tab. 10. First, we find that $\mathcal{L}_{func'-L2}^k$ is generally not useful. Moreover, when independently used, $\mathcal{L}_{func-L2}$ brings more improvement than $\mathcal{L}_{func-KL}$ though it is better to combine them both.

# H    COMPLEXITY ANALYSIS AND EFFICIENT DESIGNS

In this section, we start with the complexity analysis of FCFD. The analysis suggests some directions to lower the training cost of FCFD, and we thus experimentally explore these directions in the rest of this section. We hope that the contents in this section could guide engineers to customize their implementation of FCFD if the training cost is a concern.

Table 9: Validate the effectiveness of FCFD in the context of online knowledge distillation. Both teacher and student models are trained from scratch and learn mutually from each other. Top-1 test accuracy (%) is reported. We use 't' and 's' to denote teacher and student models, respectively. **Bold** denotes the best result.

| Teacher | WRN-40-2 | WRN-40-2 | resnet56 | resnet110 | resnet32x4 | vgg13 | |
| Student | WRN-16-2 | WRN-40-1 | resnet20 | ResNet32 | resnet8x4 | vgg8 | |
|---|---|---|---|---|---|---|---|
| Vanilla | 75.61 | 75.61 | 72.34 | 74.31 | 79.42 | 74.64 | |
| DML | 77,96 | 77.88 | 74.68 | 75.47 | 79.68 | 75.83 | t |
| CRD | 78.01 | 77.39 | 73.86 | 75.53 | 79.36 | **77.23** | |
| FCFD | **78.95** | **78.65** | **75.13** | **77.4** | **80.21** | 76.62 | |
| Vanilla | 73.26 | 71.98 | 69.06 | 71.14 | 72.5 | 70.36 | |
| DML | 75.24 | 73.69 | 70.99 | 72.4 | 74.23 | 72.55 | s |
| CRD | 75.89 | 74.12 | 70.9 | 73.07 | 75.34 | 74.08 | |
| FCFD | **76.44** | **75.01** | **72.01** | **74.22** | **76.7** | **74.65** | |

Table 10: Detailed ablation over the design of $\mathcal{L}_{func}$ and $\mathcal{L}_{func'}$. Teacher: ResNet32x4. Dataset: CIFAR-100. Top-1 accuracy (%) reported.

| $\mathcal{L}_{kd} + \mathcal{L}_{task} + \mathcal{L}_{app}$ | $\mathcal{L}_{func-KL}$ | $\mathcal{L}_{func-L2}$ | $\mathcal{L}_{func'}$ | $\mathcal{L}_{func'-L2}$ | ResNet8x4 | ShuffleV1 | ShuffleV2 |
|---|---|---|---|---|---|---|---|
| ✓ | ✓ | | | | 76.15 | 77.46 | 77.85 |
| ✓ | | ✓ | | | 76.43 | 77.94 | 77.84 |
| ✓ | ✓ | ✓ | | | 76.32 | 77.92 | 78.02 |
| ✓ | | | ✓ | | 76.62 | 77.19 | 77.43 |
| ✓ | | | | ✓ | 75.58 | 76.56 | 76.30 |
| ✓ | ✓ | ✓ | ✓ | | 76.62 | 78.12 | 78.18 |
| ✓ | ✓ | ✓ | ✓ | ✓ | 76.57 | 77.82 | 78.14 |

### H.1 COMPLEXITY

Suppose that both teacher and student have $N$ stages. Given a path used for feature matching $< k, \delta >$, the triggered additional training cost is:

$$Additional\ Training\ Cost = \begin{cases} cost(B_{ts}^k) + \sum_{l=k+1}^{N} cost(M_s^l) + cost(C_s), & \delta = 0 \\ cost(B_{st}^k) + \sum_{l=k+1}^{N} cost(M_t^l) + cost(C_t), & \delta = 1 \end{cases} \quad (21)$$

Comparing with existing methods, our FCFD involves the specially cost terms $\sum_{l=k+1}^{N} cost(M_s^l)$ and $\sum_{l=k+1}^{N} cost(M_t^l)$. However, FCFD only needs very simple bridge modules and does not incorporate complex mechanisms like contrastive learning (Tian et al., 2020; Xu et al., 2020), which lowers the overall complexity of FCFD.

There are mainly two factors that determine the practical cost of FCFD: the number of paths sampled in every iteration, $|\mathbb{S}|$, and from which stage we start to do functional matching, namely the least value of $k$ in $< k, \delta >$, and we denote this as $k_{min}$. Given a path $< k, \delta >$, the smaller $k$ is, the more extra blocks will be passed through, so larger $k_{min}$ lowers the overall extra cost.

After dividing the networks into 4 stages, there are 4 candidate positions for features matching: $F^1$, $F^2$, $F^3$, and $F^4$. In practice, we set $k_{min} = 2$, and we also do not functionally match the last feature, $F^4$, since the focus of FCFD is to investigate the effectiveness of functional matching over intermediate features. There are thus four candidate paths: $\{< k = 2, \delta = 0 >, < k = 3, \delta = 0 >, < k = 2, \delta = 1 >, < k = 3, \delta = 1 >\}$. In each iteration, as mentioned in Sec. 2.3, we by default sample two paths from the candidates.

Now we empirically compare the training speed and memory cost of FCFD with existing methods. We choose the ResNet8x4-ResNet32x4 pair on CIFAR100. Experiments are conducted on the same machine with irrelevant variables aligned. Results are shown in Tab. 11. DKD (Zhao et al., 2022) is the least complex method and its complexity is by theory exactly identical to the original KD (Hinton et al., 2015). Meanwhile, our FCFD is faster than SOTA feature distillation method

Table 11: Training complexity. Teacher: ResNet32x4. Student: ResNet8x4. Dataset: CIFAR-100.

| Method | DKD | Review | CRD | FCFD |
|---|---|---|---|---|
| Seconds / Epoch | 33.12 | 48.40 | 54.02 | 43.98 |
| Peak GPU Memory Usage (MB) | 459 | 1231 | 790 | 1132 |
| Top-1 Acc (%) | 76.32 | 75.63 | 75.56 | 76.62 |

Table 12: Effect of different number of sampled paths ($|\mathbb{S}|$) for functional feature matching. Teacher: ResNet32x4. Dataset: CIFAR-100. Top-1 accuracy (%) is reported for all student models, and training cost is reported for ResNet8x4 for reference.

| $|\mathbb{S}|$ | ShuffleV1 Top-1 | ShuffleV2 Top-1 | ResNet8x4 Top-1 | ResNet8x4 Seconds / Epoch | ResNet8x4 Peak Memory (MB) |
|---|---|---|---|---|---|
| 1 | 77.50 | 77.69 | 75.92 | 36.76 | 868 |
| 2 | 78.12 | 78.18 | 76.62 | 43.98 | 1132 |
| 4 | 78.20 | 78.98 | 77.07 | 61.27 | 1268 |

Review (Chen et al., 2021) and CRD (Tian et al., 2020). The results show that training complexity is not a significant shortage of FCFD.

## H.2 Effect of number of sampled paths $|\mathbb{S}|$

We show in Tab. 12 the effect of using a different number of sample paths per iteration. A clear trend is that more sampled paths improve the accuracy while inducing longer training time and larger memory costs. We find that with all 4 candidate paths selected in one iteration, the final accuracy of the student model can be very high. Therefore, for scenarios where model accuracy is of great importance while computing resources are relatively ample, we recommend directly setting larger $|\mathbb{S}|$. When resources are limited, $|\mathbb{S}|$ could be lowered accordingly.

## H.3 Effect of different matching positions

As mentioned before, it brings more training costs to functionally match shallow features. As a result, starting function matching since a relatively deep position (*i.e.*, increasing $k_{min}$) can lower the cost of FCFD. We conduct experiments to explore how the performance of the student changes as we select different positions for functional feature matching, the results are shown in Tab. 13.

The conclusion is that, while functional matching is effective over both shallow and deep features, it brings larger improvement when applied only to deep features than only to shallow features. Meanwhile, involving both shallow and deep features with random sampling is of medium speed, and performs the best. Taking the $|\mathbb{S}| = 4$ case in Tab. 12 into consideration, we can conclude that the advantage from functionally matching shallow and deep features can be accumulated.

## H.4 Effect of using partial terms in $\mathcal{L}_{func}$

As shown in Eq. 6, for calculating $L_{func}$, we make $B_{st}^k(F_s^k)$ pass through all the later teacher modules ($l \in [k+1, N+1]$). Another possible approach to lower the training cost of FCFD is to only propagate $B_{st}^k(F_s^k)$ through the first several teacher modules. Since in our scenario a transformed student feature at most passes through 2 teacher modules (classifier $C_t$ excluded), we conduct an experiment to see how the student performs when $L_{func}$ only involves one-module propagation, namely:

$$\mathcal{L}_{func-partial}^k = L2\left( M_t^{k+1}(F_t^k), M_t^{k+1}\left( B_{st}^k\left( F_s^k \right) \right) \right). \tag{22}$$

The result is shown in Tab. 14. First, replacing $\mathcal{L}_{func}^k$ with $\mathcal{L}_{func-partial}^k$ is an effective strategy as most of the improvement is preserved after the replacement. However, compared with the *functionally match $F^3$ only* strategy in Tab. 13, the partial-propagation strategy achieves lower accuracy

Table 13: Effect of conducting functional matching at different positions. $|\mathbb{S}|$ is 2 for all experiments. Teacher: ResNet32x4. Dataset: CIFAR-100. Top-1 accuracy (%) is reported for all student models, and training cost is reported for ResNet8x4 for reference.

| Setting | ShuflfleV1 Top-1 | ShuffleV2 Top-1 | ResNet8x4 | | |
|---|---|---|---|---|---|
| | | | Top-1 | Seconds / Epoch | Peak Memory (MB) |
| Functionally match $F^2$ only | 77.03 | 77.78 | 76.46 | 49.38 | 1132 |
| Functionally match $F^3$ only | 77.75 | 77.90 | 76.60 | 38.73 | 891 |
| Functionally match $F^2$ and $F^3$ | 78.12 | 78.18 | 76.62 | 43.98 | 1132 |

Table 14: Replacing $\mathcal{L}^k_{func}$ with $\mathcal{L}^k_{func-partial}$. Teacher: ResNet32x4. Dataset: CIFAR-100. Top-1 accuracy (%) is reported for all student models, and training cost is reported for ResNet8x4.

| Setting | ShuflfleV1 Top1 | ShuffleV2 Top1 | ResNet32x4 | | |
|---|---|---|---|---|---|
| | | | Top1 | Seconds/Epoch | Peak Memory (MB) |
| with $\mathcal{L}^k_{func}$ | 78.12 | 78.18 | 76.62 | 43.98 | 1132 |
| with $\mathcal{L}^k_{func-partial}$ | 77.69 | 77.72 | 76.52 | 37.13 | 980 |

while the training cost is almost similar. Therefore, in practice, if the computing resources are really scarce, it is suggested to try raising $k_{min}$ first before trying $\mathcal{L}_{func-partial}$.

## I    DIFFERENCES BETWEEN FCFD AND EXISTING WORKS

Some existing works (Bai et al., 2020; Li et al., 2020a;b; Yang et al., 2021) also feed representations from one network to another network to distill knowledge. In this section, we analyze their differences from FCFD.

From the perspective of problems to solve, *Bai et al. (2020) and Li et al. (2020a) are designed to solve problems that only exist in special topics and do not generalize to common KD scenarios.* Specifically, Bai et al. (2020) aims to alleviate overfitting in few-shot distillation, so they propose a block-wise supervision strategy. Li et al. (2020a) focuses on a more special scenario, where a residual network helps a non-residual network to overcome gradient vanishing. In contrast, FCFD is motivated by the function consistency problem that widely exists in feature distillation scenarios, and is applicable and effective in general knowledge distillation cases. *From the methodology perspective*, there are still clear differences: Bai et al. (2020) directly matches the behavior of paired teacher and student modules, so they feed the same input feature to the two blocks and hope the output could be similar. In contrast, FCFD takes a feature-matching perspective and hopes paired features could lead to similar outputs after process by the same module. Our method can deal with situations where teacher and student features have different shapes, while Bai et al. (2020) cannot. For Li et al. (2020a), though they do put student features into the teacher network, their aim is to obtain the residual gradient which is originally inaccessible for a non-residual network. Furthermore, they do not feed teacher features into the student modules, which we show to be effective with ablation. While SRRL (Yang et al., 2021) does target on the general KD scenario, it focuses on a narrower problem of penultimate feature matching and can be considered as a special case of FCFD, where only penultimate features are functionally aligned with $\mathcal{L}_{func}$. The comparative experiment in Tab. 2 also shows that FCFD clearly outperforms SRRL.

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
