# OpenReview forum: "Function-Consistent Feature Distillation"
_ICLR.cc/2023/Conference — ICLR 2023 poster_

### Official Review · Reviewer_8x8h · 2022-10-23

**Confidence:** 4
**Correctness:** 3
**Technical Novelty And Significance:** 3
**Empirical Novelty And Significance:** Not applicable
**Recommendation:** 8

**Clarity, Quality, Novelty And Reproducibility:**

Clarity: 8/10

Quality: 8/10

Novelty: 6/10

Reproducibility: Unknown

**Strength And Weaknesses:**

1. The asymmetric loss design for the student and teacher network requires more ablation studies. Specifically, when “using student’s lateral part as the lens”, the authors think that “randomly initialized bridge” may bring unstable training status, and thus feature-level supervision is not available in "$L_{func’}^{k}$". When considering “teacher’s lateral part”, the bridge module still exists, however, the feature-level supervision is activated in "$L_{func}^{k}$". There had better be ablations like:
    1. Only KL supervision in "$L_{func}^{k}$"
    2. KL plus L2 supervision in "$L_{func’}^{k}$"

    Moreover, the following ablations are also expected:
    1. Only L2 supervision in "$L_{func}^{k}$"
    2. Only L2 supervision in "$L_{func’}^{k}$"
    3. L2 supervision on partial layers (Currently, all layers after the "k-th" one seem to be considered in the loss. Is it the best practice?)

2. For the path sampling scheme used in this paper, the authors simply sample two paths during training. I wonder the effects of different path sampling schemes, such as the number of sampled paths, the sampling probability (uniformly or imposing some priors to different layers) and the relative number of teacher-to-student and student-to-teacher paths if there are more than two paths. By the way, the sampling seems to be canceled for object detection, it might also work for classification to freeze the sampled path during the whole training process. In short, I think more ablations about the sampling scheme should be included in this paper.

3. Although the authors have repeatedly emphasized the extra training cost is acceptable, I think some numerical indicator, such as the training time and GPU memory peak, should be included in the experimental results.

4. Can the proposed method work for segmentation? Intuitively, I still think the propagation of sampled paths would introduce a lot of cost, which might hinder the training of dense prediction tasks, such as semantic segmentation.

5. For the separate batch statistics, which one will be applied during inference, or is there some fusion scheme to mix statistics of different paths during inference?

6. I observe that “FCFD+DKD” is inferior to “FCFD” under the settings of “WRN-40-2/WRN-40-1” and “WRN-40-2/WRN-16-2” in Table 4. Still in Table 4, the result of “SimKD” under “WRN-40-2/WRN-16-2” is missing, and “FCFD+SimKD” is also inferior to “FCFD” under this setting. Is there an explanation for these results?

7. Can the authors provide some feature visualizations to explain the results in Table 6? For example, the PCA of student and teacher features w.r.t. “FCFD” and "$L_{kd}+L_{task}+L_{app}$".


**Summary Of The Paper:**

This paper proposes a new knowledge distillation (KD) method to better align student features with teacher features. The authors claim that traditional L2 loss for feature-level KD is not appropriate due to the deviation of functional mapping of similar features. In order to alleviate this problem, a function-consistent feature-level KD is utilized to align the cross-propagated features of both the student and the teacher. Various designs, such as separate batch statistics and path sampling, are incorporated to help the training and inference process. Experiments on both classification and object detection datasets demonstrate the effectiveness of the proposed method.

**Summary Of The Review:**

Overall, this paper could be accepted if the authors can answer the aforementioned concerns to some extent.

---

> ### Author Response · Authors · 2022-11-11
> **Reply to Reviewer 8x8h -- part1**
>
> First of all, we'd like to thank you for your constructive suggestions! **All experiments you referred to are now updated in **Appendix E (detailed ablations over loss design) and F (complexity analysis and efficient designs)** in the revision**, and we strongly recommend you to read these newly-added sections for details.
>
> # About the asymmetric loss design
>
> We'd like to first make a clarification that feature level matching for $\mathcal{L}\_{func}$ and $\mathcal{L}\_{func'}$ really face different situations.  For $\mathcal{L}\_{func}$, feature level matching is achieved by $\textbf{d}\_{fd}\Bigg({M\_t^{l} \circ \cdots \circ M\_t^{k+1}\left(F\_t^k\right)}, M_t^{l} \circ \cdots \circ M\_t^{k+1}\Big( B\_{st}^k\left(F\_s^k\right)\Big)\Bigg)$. You can see that the first term, ${M\_t^{l} \circ \cdots \circ M\_t^{k+1}\left(F\_t^k\right)}$, does *not* involve any randomly initialized blocks that need optimization, so we are making a random feature (the second term) mimic a stable, well-trained feature (the first term). However, in the  $\mathcal{L}_{func'}$ case, if we do feature level matching, it will be like this: $\textbf{d}\_{fd}\Bigg({M_s^{l} \circ \cdots \circ M_s^{k+1}\left(B\_{ts}^k (F\_t^k)\right)}, M_s^{l} \circ \cdots \circ M\_s^{k+1} (F\_s^k)\Bigg)$. You can see that due to $B\_{ts}^k$ and $M\_s^{(\cdot)}$ , both of the two features are random and need optimization, which hardly forms effective constraint for the student model. Our experiments also show that penalizing their difference has little effect. (**Tab. 8 in Appendix**)
>
>
>
> # More ablations
>
> The ablations you mentioned are now in **Appendix E**, which shows that our loss design is generally optimal. The *L2 supervision on partial layers* experiment is discussed in **Appendix F.4** as an option to make training more efficient, and the results show that while this could raise the speed of FCFD while preserving most of the accuracy improvement, a different strategy, which only functionally matches deeper layers, turns out to perform better with similar training cost.
>
> # Path sampling strategy
>
> The effect of sampling different number of paths per iteration is now in **Appendix F.2**. The overall conclusion is that more sampled paths increases the performance of the student while adding extra training cost.  We agree that exploring the effect of even more complicated path sampling strategy is an interesting topic, but since it could be complex and is not closely related to the core message that we want to deliver, we consider it as future work.
>
> # Training Cost
>
> Following your suggestion, **Appendix F** now presents a detailed analysis over the topic of training cost. The training is actually faster with FCFD than the well-known method CRD[1] and Review [2].  To train an epoch with resnet8x4-resnet32x4, FCFD spends 43.98s, while 48.40s and 54.02s for Review and CRD. FCFD also leads to less peak memory usage than Review. **Appendix F** also involves experiments and discussions over  designs that make FCFD more efficient.
>
> # BN Statistics
>
> The statistics accumulated for the pure student path will be used for inference. For example, in Fig.3, it is the statistics with blue shadow and marked with $M^2_t\circ M^1_t$ that will be used.
>
>
> #　Apply FCFD to Segmentation
>
> As we analyze in Appendix F.1, the additional training cost of FCFD is closely related to the position of features that we want to functionally match. Deeper features correspond to less lateral blocks. Therefore, just as what we have done on object detection, where we only functionally match the features output by FPN so the lateral modules are only ROI heads, we can also apply FCFD to semantic segmentation with properly designed matching position.  For example, consider the deeplabv3+ pipeline:
>
> >```
> >f1, f2, f3, f4 = self.backbone(x)
> >aspp_feat = self.aspp(f4)
> >x = self.decoder(aspp_feat, f2)
> >```
>
> Then as an efficient implementation of FCFD, we can first let input $x$ pass though both the backbone of teacher and student independently without considering functional matching. Then we can consider that the function of f2 and f4 are now to serve as the input to the lateral aspp and decoder modules, so we can functionally match f2 and f4 by cross propagate them into aspp and decoder modules, and match the intermediate and final results. In this way, with resnet101 as teacher and use Pascal-VOC  as dataset, we have raised the mIOU of a resnet18 from 72.07 to 73.66.  Note that due to limited time we have yet to tune the hyperparameters so it is hard to say how large its maximum improvement could be, but at least the experiments make it safe to conclude FCFD is *applicable* on dense prediction tasks.
>
>
>
>
> [1] Contrastive representation distillation, ICLR 2020
>
> [2] Distilling knowledge via knowledge review, CVPR 2021.

---

> ### Author Response · Authors · 2022-11-11
> **Reply to Reviewer 8x8h -- part2**
>
> # Results in Table4
>
> **SimKD WRN-40-2/WRN-16-2 result**
>
> The DKD and SimKD results are directly borrowed from their original paper.  Since the result of WRN-40-2/WRN-16-2 experiment is not reported by original authors, we left it blank. As suggested by reviewer co3V, in the revision we have also added our reimplemented results into the table.
>
> **Why FCFD+SimKD is inferior to FCFD**
>
> Actually we think the explanation of this phenomenon has been covered in the last paragraph of Sec. 3.2:
>
> > The improvement is much larger when the teacher is ResNet32x4 rather than WRN-40-2, **one possible explanation is that the performance of WRN-40-2 itself is relatively low and has barricaded the progress of the student**. As shown in Tab. 4, with either WRN-16-2 or WRN-40-1 as student, the performance of the student has approached or even surpassed the teacher.
>
> We think the performance of wrn-16-2 is likely to have saturated in the setting.
>
> # Visualization for feature similarity
>
> We'll do it in the next few days and we will update the results when we are finished. (**update: visualization for feature similarity is now shown in Appendix. H**)

---

> ### Author Response · Authors · 2022-11-17
> **Sincerely expect your further comments**
>
> Dear Reviewer 8x8h,
>
> Considering the deadline for discussion stage 1 is approaching, we sincerely ask if our response has addressed your concerns. If not, could you please list your follow-up questions? We are happy to have a further discussion with you. Thanks!
>
> Bests, Paper311 authors

---

### Official Review · Reviewer_co3V · 2022-10-24

**Confidence:** 4
**Correctness:** 3
**Technical Novelty And Significance:** 2
**Empirical Novelty And Significance:** Not applicable
**Recommendation:** 5

**Clarity, Quality, Novelty And Reproducibility:**

The presentation of this paper is basically clear. My main concerns are to the limited technical contribution/novelty, and unfair experimental comparisons. Besides, the implementation of FCFD may not easy as there are many hyper-parameters. No code is provided.

Please refer to my comments in 'Strength And Weaknesses' for details.

**Strength And Weaknesses:**

Strengths.

+ Improving feature matching is a critical research topic in FD research.

+ The basic ideas of the proposed method are easy to understand.

+ Experimental comparisons are performed on both image classification and object detection tasks with diverse teacher-student network pairs.

Weaknesses.

- The method.

The novelty of the proposed method, FCFD, is very limited. Note that the core designs of FCFD are two kinds of lateral network parameter reuses: feeding dimension-aligned student features at a certain layer to the paired teacher layer, and feeding dimension-aligned teacher features at a certain layer to the paired student layer. However, these two ideas have already been well explored in many existing knowledge distillation works such as cross distillation [1], residual distillation [2], explicit connection distillation [3],  softmax regression representation learning [4]. Unfortunately, these works are completely missed by the authors. Given the existence of these works, FCFD has no new technical contribution, to the best of my knowledge.

[1] Few Shot Network Compression via Cross Distillation, AAAI 2020.

[2] Residual Distillation: Towards Portable Deep Neural Networks without Shortcuts, NeurIPS 2020.

[3] Explicit Connection Distillation, ICLR 2020 submission.

[4] Knowledge distillation via softmax regression representation learning, ICLR 2021.

- The experiments.

Experimental comparisons are not convincing enough.

For experimental comparisons on image classification datasets (CIFAR100 and ImageNet), in Table 4, etc., it seems that the authors directly use the results reported in the papers of Review KD and DKD as the baselines. This is not fair, as for ablation, the comparison of FCFD+DKD/FCFD+SimKD vs. DKD/SimKD needs to run on the same training machines and code settings.

For experimental comparisons on object detection dataset MS-COCO, experimental settings are also not optimal: 1) for object detection task, mainstream feature distillation methods, such as [1-3], already reuse/share feature pyramid network (FPN, the neck) of the teacher backbone as the FPN for the student. In this context, is it necessary to apply the proposed FCFD? 2) For experimental comparisons, FCFD should be compared to mainstream feature distillation methods for object detection, but not FD methods for image classification as they usually perform much worse.

In the final formulation of FCFD, there are five loss terms, how to set proper weights to them?

How about the training cost of FCFD? As it will lead to heavy extra memory cost, it is necessary to compare training cost of FCFD with other FD methods.

[1] Distilling Object Detectors with Feature Richness, NeurIPS 2021.

[2] Instance-Conditional Knowledge Distillation for Object Detection, NeurIPS 2021.

[3] Focal and global knowledge distillation for detectors, CVPR 2022.

**----Update----**

Balancing both positive and negative aspects of the rebuttal from the authors, although I raised the score from 3 to 5 (I would more like to give a score of 4 if there is such a choice), I still think this paper is not good enough to reach the acceptance bar of ICLR.

**Summary Of The Paper:**

This paper explores the new way to measure the distance between the teacher and student features, for better feature distillation (FD) performance. Specifically, the authors first point out the commonly used L2-norm feature distance is not good in semantic function, and then present a new FD method called function-consistent feature distillation (FCFD). Besides logits based KD loss at the heads and common FD loss at some intermediate teacher-student layer pairs, FCFD introduces two extra loss terms based on two kinds of lateral network parameter reuses: feeding dimension-aligned student features at a certain layer to the paired teacher layer, and feeding dimension-aligned teacher features at a certain layer to the paired student layer. The effectiveness of the proposed method is validated on both image classification (CIFAR100 and ImageNet) and object detection (MS-COCO) tasks.

**Summary Of The Review:**

The novelty of this paper is very limited.

Please refer to my comments in 'Strength And Weaknesses' for details.

---

> ### Author Response · Authors · 2022-11-11
> **The Novelty**
>
> A comparison between FCFD and [1-3] is now added in the Appendix. From the perspective of problems to solve, *[1] [2]  are designed to solve problems that only exist in special topics and do not generalize to common KD scenarios*. Specifically,  cross distillation [1] aims to alleviate overfitting in few-shot distillation, so they propose a block-wise supervision strategy. Residual distillation [2] focuses on a more special scenario, where a residual network helps a non-residual network to overcome gradient vanishing. In contrast, FCFD is motivated by the function consistency problem that widely exists in feature distillation scenarios, and is applicable and effective in general knowledge distillation cases. *From the methodology perspective*, there are still clear differences: cross distillation[1] directly matches the behavior of paired teacher and student modules, so they feed the same input feature to the two blocks and hope the output could be similar. In contrast, FCFD takes a feature-matching perspective, and hopes paired features could lead to similar outputs after process by the same module. Our method can deal with situations where teacher and student feature have different shapes, while cross distillation cannot. For residual distillation [2], though they do put student features into teacher network, their aim is to obtain the residual gradient which is originally inaccessible for a non-residual network. Furthermore, they do not feed teacher features into the student modules, which we show to be effective with ablation.  While [3] does target on the general KD scenario, it can be considered as a special case of FCFD, where only penultimate features are functionally aligned with $\mathcal{L}_{func}$. Comparative experiment with [3] is also appended into Tab.2, which clearly shows the advantage of FCFD. ECD[4] falls into the online knowledge distillation category and makes the student serve as a part of a large model ensemble. Overall, the similarity between FCFD and ECD [4] is limited.
>
> Actually, the operation of propagating features through different networks itself is relatively simple idea, and it is not surprising that relevant designs have already been explored. However, we believe that this direction is far from *well* explored. As we show in the paper, from a practical perspective, the performance of FCFD is clearly better than existing feature distillation methods while its training cost is actually not high. But before FCFD, no work has even indicated that methods in this direction has such potential in the general KD scenario. Therefore, we believe the technical contribution of FCFD *is* important. From a methodology perspective, though some design elements appeared in existing works, they were designed for different motivation and purpose, and the overall methodology of FCFD still differs from any of the existing works. Moreover, in the revision, we have added more analysis over the impact of design choices on performance, which further improves the practical and technical value of FCFD.
>
> *More importantly, we think the contribution of FCFD is more than its specific technical design. The insight that feature matching should take how features are used into consideration, which is the guideline of the methodology, is more important.* Again, this methodology is not complex (as some of the components have even been explored before for solving other tasks), but the reason why work with similar overall design and performance in general KD did not appear before FCFD is very likely since there was no such insight to motivate the overall design.  We believe the insight is important because 1) it does exist, as we analyze with toy example and empirical analysis and 2) following this insight, methods really work. We also believe that such insight can inspire researchers to devise new methods on different tasks in the future because the detection-version FCFD is exactly such a case. The advantage of FCFD on both classification and detection shows the general significance of the insight of function consistency.
>
> [1] Few Shot Network Compression via Cross Distillation, AAAI 2020.
>
> [2] Residual Distillation: Towards Portable Deep Neural Networks without Shortcuts, NeurIPS 2020.
>
> [3] Knowledge distillation via softmax regression representation learning, ICLR 2021.
>
> [4] Explicit Connection Distillation, ICLR 2020 submission.

---

> ### Author Response · Authors · 2022-11-11
> **Experiments**
>
> # Classification Experiments:
>
> ### Directly quoting existing results
>
> For Tab.4, we agree with your that reporting our reimplemented results of DKD and SimKD makes conclusions more persuasive, so we have added them in the revision.
>
> **For all other comparative experiments, we are confused why it is not fair to quote the results reported by original authors**. We are following a popular public benchmark established by CRD[2]. The training settings, including number of training epochs, data augmentation, selection of  optimizer and optimizer hyper-parameters, are all clearly regulated by the benchmark and we strictly obey. Furthermore, the pre-trained teacher's checkpoints can also be downloaded from CRD's official github repo[1], so the teacher model is also exactly matched. Our implementation of backbone model architecture is also copied from [1]. Therefore everything that is method-agnostic is the same. We note that it is not like object detection that the accuracy of the same method differs largely from framework to framework (eg. detectron2 and mmdetection), for classification the implementation is simple so it is easy to align task-agnostic implementations.
>
> We also note that quoting existing results is the standard practice in the KD community as well as many other areas.  ReviewKD quoted the results from CRD, DKD quoted the results from ReviewKD, and we quoted the results in DKD. The work SRRL[8], which you mentioned, also directly quotes the results from CRD. If you think we are missing something or not well understand your questions, please let us know.
>
> # Detection Experiments:
>
> ### Is FCFD compatible with interiting strategy?
>
> To the best of knowledge, the FPN/HEAD inheriting strategy is proposed as an parameter *initialization* method [5] [6]. After training begins the teacher and student still have different parameters. So inheriting strategy does not change the applicability of FCFD.
>
> ### The settings are not optimal
>
> **Generally speaking, we don't agree with your statement that our experiment settings are not optimal**. The reasons are as follows:
>
> 1) **Our proposed FCFD is a general knowledge distillation framework, so we believe it is necessary and very important to compare with existing methods that also claim such generality.**
>
> 2) **We also don't agree with your statement that generic feature distillation methods usually performs *much worse* than those specially designed for object detection**.  For example, the  detection-specialized distillation method ICD[5], and the generic distillation method ReviewKD [7], were both published in 2021, and luckily they both tested on the Faster-RCNN-FPN R50-R101 1x setting. The mAP of ReviewKD is 40.36% while the mAP of ICD is 40.4% without inheriting strategy, which is almost the same. Both of the two methods are implemented with detectron2, and they do use the same pre-trained teacher and student checkpoint, so the results should be comparable. As a result, by comparing with works in Tab.3, **we are not comparing with weak baselines**.
>
> 3) **FCFD outperforms the mainstream detection-specialized method ICD that you list**. We have run the MV2-R50 experiment with ICD[5], with training settings consistent with our experiments (including loaded checkpoints).  The mAP is 32.88, which is significantly lower than FCFD (34.97). We are still working on the R18-R101 setting and we will update all the results when experiments are done. (**update: ICD experiments are all done and results have been updated in Tab. 3**)
>
> 4) The other two works that you mentioned, FGD[4] and FRS[5], are implemented on mmdetection, while all results in our Tab. 3 are obtained with detectron2, so it is hard to do a fair comparison (e.g. FGD[4] did not compare itself with ICD[5] though cited it). We believe current results are adequate to prove the effectiveness of FCFD on detection.
>
> By the way, there *is* actually already one method specially designed for detectors in Tab.3, namely FGFI[3].
>
>
>
> [1] https://github.com/HobbitLong/RepDistiller
>
> [2] Contrastive representation distillation, ICLR 2020
>
> [3] Distilling object detectors with fine-grained feature imitation, CVPR 2019
>
> [4] Distilling Object Detectors with Feature Richness, NeurIPS 2021.
>
> [5] Instance-Conditional Knowledge Distillation for Object Detection, NeurIPS 2021.
>
> [6] Focal and global knowledge distillation for detectors, CVPR 2022.
>
> [7] Distilling knowledge via knowledge review, CVPR 2021.
>
> [8] Knowledge distillation via softmax regression representation learning, ICLR 2021

---

> ### Author Response · Authors · 2022-11-11
> **Training cost & hyperparameters**
>
> Thank you for your comments and questions!
>
> # Training Cost
>
> Even though the training cost of FCFD may seem to be large, the actual cost is not larger than well-known feature distillation methods, e.g. CRD[1] and Review [2]. To train an epoch with resnet8x4-resnet32x4, FCFD spends 43.98s, while 48.40s and 54.02s for Review and CRD. FCFD also leads to less peak memory usage than review. A detailed analysis of training cost is now updated in the revision (**Appendix F**). We note that the following points make the cost acceptable:
>
> 1. We only functionally match relatively deep features, so the *lateral part of the network* is not that large.
> 2. We use random sampling strategy to lower the overall training cost.
> 3. We use very light bridge module and do not incorporate extra modules for complex feature processing.
>
> Besides comparisons with SOTA KD methods over training speed and peak GPU memory usage, in Appendix F we also discuss some techniques that makes FCFD more efficient. We agree that this complexity analysis section is very important and we apologize for its absence in the last version.
>
> [1] Contrastive representation distillation, ICLR 2020
>
> [2] Distilling knowledge via knowledge review, CVPR 2021
>
> # hyper-parameters
>
> The selection of hyper-parameters are now appended in **Appendix A**. There are actually not that many hyper-parameters to tune because we make many loss terms share the same weight. Specifically, the weights for $\mathcal{L}\_{task}$, $\mathcal{L}\_{kd}$, $\mathcal{L}\_{func'}$, and the KL-Divergence term in $\mathcal{L}\_{func}$, are always the same and set to 1 in all our experiments. $\mathcal{L}\_{app}$ and the $L2$ terms in $\mathcal{L}\_{func}$ also always share the same weight factor, which is selected from two candidates: 0.02 and 5.0. Similarly, for $\mathcal{L}\_{kd}$, $\mathcal{L}\_{func}$, and the KL term in $\mathcal{L}\_{func'}$, the same temperature is used, and is selected to be either 4 or 8. Therefore, there are merely 3 hyperparameters involved and only 2 of them need tuning. We think this is not complex comparing with existing methods.

---

> ### Author Response · Authors · 2022-11-17
> **Sincerely expect your further comments**
>
> Dear Reviewer co3V,
>
> Considering the deadline for discussion stage 1 is approaching, we sincerely ask if our response has addressed your concerns. If not, could you please list your follow-up questions? We are happy to have a further discussion with you. Thanks!
>
> Bests, Paper311 authors

---

### Official Review · Reviewer_pBzA · 2022-10-25

**Confidence:** 3
**Correctness:** 4
**Technical Novelty And Significance:** 3
**Empirical Novelty And Significance:** 3
**Recommendation:** 8

**Clarity, Quality, Novelty And Reproducibility:**

The writing is clear, the whole paper is easy to understand. The novelty seems sufficient to me. The experiments are all conducted on public benchmarks.

**Strength And Weaknesses:**

strength:
+ A novel distillation method, FCFD, is proposed emphasizing the matching between teacher and student intermediate features not w.r.t pre-defined losses like L2 but w.r.t. function (aka the lateral network).
+ Extensive experiments on image classification and object detection are conducted to investigate the effectiveness and generalization of FCFD. The results show that the proposed FCFD significantly outperforms state-of-the-art knowledge distillation methods.
+ The proposed FCFD is compatible with many advanced knowledge distillation techniques with orthogonal contributions. Experiments verify its effectiveness when combined with other methods.
+ I like the toy example given in Figure 1, it illustrate some limitation of the L2 loss. (I suggest also mentioning other losses like smooth L1).

weakness:
- Even though the random sampler is introduced in Section 2.3, I could not find how it is implemented in experiments. Especially how the \delta values be sampled?
- In Table 4, why SimKD result under WRN-40-2/WRN-16-2 is not given? The values in this row seem to be directly borrowed from the SimKD paper, but without a confidence interval. If the authors reproduced SimKD results, then why not filling all the cells?


**Summary Of The Paper:**

This paper proposes a novel feature distillation method. Instead of directly restricting features between teacher and student using explicit loss functions like L2, the proposed method, FCFD, utilizes lateral networks to define the feature loss,  and improves over existing feature distillation methods.

**Summary Of The Review:**

The idea of distilling using a part of the network seems novel to me. Extensive experiments have been conducted to verify the effectiveness of the proposed method in both image classification and object detection. Moreover, the authors illustrate that the proposed method can be effectively combined with other existing methods, showing a strong potential in application. In general, I recommend acceptance of this paper.

---

> ### Author Response · Authors · 2022-11-11
> **Thank your for your comments!**
>
> Thank you for your appreciation of the insight of FCFD!
>
> ### The implementation of random sampling:
>
> In brief, the sampling of $k$ and $\delta$ in $\mathbb{S}$ is independent. Specifically, after dividing the model into 4 stages,  there are 4 candidate positions for features matching: $F^1$, $F^2$, $F^3$, $F^4$. In practice we only functionally align $F^2$, $F^3$. There are thus 4 candidate paths: {<k=2,$\delta$=0>,<k=3,$\delta$=0>,<k=2,$\delta$=1>,<k=3,$\delta$=1>}. In each iteration we randomly sample two paths following a uniform distribution.
>
> As suggested by reviewer8x8h, we have also added some experiments and analysis over the sampling strategy into the revision (Appendix F.2). Sampling more paths per iteration will make accuracy higher but also inducing more training cost.
>
> ### WRN-40-2/WRN-16-2 results in Table 4
>
> We indeed directly borrowed the SimKD results from their original paper. We did not report the confidence interval because the DKD paper did not report it, so for style uniformity we only report the mean. We have added the description of data source into the table captions.  As suggested by reviewer co3V, we have also added our reimplemented results into the table.
>
> Following your suggestion, in the revision, we mention that L1 and Smooth L1 suffer from the same drawback as L2. Thank you for this suggestion!
>
> Considering the deadline for discussion stage 1 is approaching, we sincerely ask if you have more questions and suggestions.  We are happy to have a further discussion with you. Thanks!

---

> > ### Comment · Reviewer_pBzA · 2022-12-07
> > **On authors' response and other reviews**
> >
> > I'm satisfied with the authors' responses, not only to my reviews but also to others. I would keep my scores.

---

> > > ### Author Response · Authors · 2022-12-11
> > > **Thank you for your feedback!**
> > >
> > > Dear Reviewer pBzA,
> > >
> > > Sincerely thank you again for your time and patience paid on our work, and the constructive comments that have helped improve the paper!
> > >
> > > Bests, Paper311 authors

---

### Official Review · Reviewer_ScN2 · 2022-10-26

**Confidence:** 4
**Correctness:** 3
**Technical Novelty And Significance:** 3
**Empirical Novelty And Significance:** 3
**Recommendation:** 5

**Clarity, Quality, Novelty And Reproducibility:**

I can understand most parts of the paper though some expressions such as ``appearance'' and ``function'' are weird. I am on the border of the novelty. Some mathematical insights or proof sketches are missing to support the author's main claims. Some key details, such as the projection module B() are not sufficiently well-described. However, I choose to trust the results of the paper.

**Strength And Weaknesses:**

Strength:
- The paper proposes a novel function-consistent feature distillation method targeting the computer vision scenario.
- The discussion on negative transfer from L2 loss between features is interesting.
- The experimental evaluation is adequate, and the results somehow support the main claims.

Weakness:
- The motivation is not clear to me. The author claimed in the paper that there are some intermediate Conv and BN modules to map the student features to teacher features into the same size. Some methods, such as Review[1] and MGD[2], all consider multiple projection layers. It seems that the negative transfer from L2 loss is curtailed by these projection modules. I wonder if there is any theoretical analysis that supports the author's claim. A toy example of shallow learning can not provide adequate support for author's claim from a deep learning perspective.
- The method is complicated. The proposed method considers both the student and teacher's lateral parts of the network, and then the training cost is relatively high. To me, the minor performance improvement can not make up for the additional complexity of the system.
- Some important experiments are missing: 1) The details of the projection module B() is missing. 2) In table 1 and table 2, the results w/o KD are omitted.

[1] Yang, Zhendong, et al. "Masked Generative Distillation." arXiv preprint arXiv:2205.01529 (2022).
[2] Chen, Pengguang, et al. "Distilling knowledge via knowledge review." Proceedings of the IEEE/CVF Conference on Computer Vision and Pattern Recognition. 2021.

I hope the authors can reply to me on the following:
- More mathematical insights on negative aspects of l2 loss.
- The complexity analysis of the proposed distillation method.
- The details on projection module B() and the missing experimental results on table 1 and table2.

**Summary Of The Paper:**

This paper introduces a function-consistent feature distillation method in the computer vision scenario. The motivation of the proposed method is based on the assumption that the widely used l2 loss between features does not take function-consistent into consideration, bearing the consequences that the student model may suffer from the negative transfer. Then the author introduces Function-Consistent Feature Distillation(FCFD), which measures the similarity between student and teacher's outputs generated by the lateral part of the same network. Extensive experiments show the improvement of the proposed method.

**Summary Of The Review:**

The paper proposes a feature distillation method targeting the computer vision scenario.
Although the author claims that the conventional l2 loss between features contains some unfixed flaws, the motivation and insight are not clear. A toy example of shallow learning is far from supporting the claim.
On the other hand, the proposed solution is complicated. Considering the improvement in performance is not significant (needless to say, some important baselines are missing), the whole cost is too high to be practical in the real industry environment.
Overall, I think the paper is below the borderline of ICLR.

---

> ### Author Response · Authors · 2022-11-11
> **More explanations about motivation**
>
> Sorry for making you confused！
>
> The most important message that we want to deliver with FCFD is the insight that function consistency should be considered for feature matching. Heartbrokenly we may have failed to convey this insight to you, but we will try our best to achieve this during this rebuttal. *We will also give you some mathematically insight from the mutual information perspective.*
>
>  *To avoid trivial misunderstandings, we'd like to first clarify that the word 'function' in FCFD means  'what something is used for' rather than "a mathematical relation"*. We also clarify that the multi-layer bridge module that you mentioned cannot help with the deficiency of L2 distance.
>
> We start with a very special case that may help you understand our motivation (it is similar to the problem consider by Overhaul[1]): Given a ReLU layer within the teacher model, it receives input feature $F_{t,pre}$ and outputs $F^t_{t,post}$, where $t$ means teacher, $pre$ and $post$ denote pre-activation and post-activation. As we all know, with ReLU, the positive values in $F_{t,pre}$ remain unchanged while the negative values will all be erased to zero. **Then what would happen if we conduct feature distillation that  makes a student feature $F_s$ to mimic $F_{t,pre}$ w.r.t. L2 distance ($Loss = L2(F_{t, pre}, B(F_s))$)?** Obviously,  differences in both positive and negative values are equally measured and penalized. This is not desirable: differences in negative values in  $F_{t,pre}$ later on changes nothing at all as long as the values are still negative,  but differences in positive values will indeed influence lateral features and the final result.  As the positive values in $F_{t,pre}$ convey most of (if not all) the information used by the teacher for solving the target task (i.e. task-relevant information), feature distillation should make the student pay more attention to positive values. But L2 distance cannot achieve this.  *Note that we do not emphasize so-called negative transfer since we believe that even in negative values in $F_{t,pre}$ there are still more or less useful cues that could hint the student. However, considering that it is impractical to make the student feature match everything in teacher feature perfectly, some priority should be attached to positive values*
>
> The above case, can be easily conquered, just by changing the loss function to $Loss = L2(ReLU(F_{t,pre}), ReLU(B(F_s)))$, in this way the student will no longer care about the negative values in $F_{t,pre}$. **This reflects the central idea of FCFD: when measuring feature similarity, we should take what the features are used for, namely the function of features, into account.** For the aforementioned case, the function of $F_{t,pre}$ is to serve as the input to a ReLU layer, so positive values should be emphasized.
>
> **We  then generalize this extreme case to the more general cases**: consider $more$ modules (denoted as $M_t'$)  located *after* a teacher feature $F_{t}'$. $M_t'$ can be as simple as a ReLU layer, a residual block, or the whole lateral teacher network after $F_{t}'$. We make $B(F_s')$ to mimic this feature. The operations within  $M_t'$  are highly complicated and of course more anisotropic than a single ReLU. Therefore, when $M_t'$  processes different input features,  differences in some aspects are more important than others. **This is validated by our experiments on CIFAR100 at the end of paragraph 2 in Sec. 1.3**. Again, just like for a pre-ReLU feature  that we want the the student to pay more attention to its positive values, here we also encourage the student to focus on aspects that more significantly influence the output. This can be achieved by defining a new loss term: $Loss = L2(M_t'(F_{t}'), M_t'(B(F_s')))$. $d(\cdot, \cdot)=L2(M_t'(\cdot), M_t'(\cdot))$ defines a new distance metric; comparing with $L2$, $d$ knows what aspects in features are important in the eye of $M_t'$, and such aspects are emphasized.
>
> Finally, what do we mean by saying "appearance" and "function"? Comparing $L2$ and $d$,  $L2$ measures the *general* similarity between two features, which feels just like judging if features $look$ alike or not, without considering how exactly the features will be used, so we call this "appearance". In contrast, $d$ considers what the features are used for, namely what are the *functions* of the features, so we call this the "function" perspective for feature matching. With appearance matching, all differences are equal; while for functional matching, the important aspects are emphasized.
>
> [1] A comprehensive overhaul of feature distillation. ICCV, 2019.

---

> ### Author Response · Authors · 2022-11-11
> **Mathematically Insight**
>
> For mathematically insight, we hope the following explanation from the mutual information perspective could satisfy you.
>
> After feeding an input image $x$ into the teacher model, it is processed in such a way: $x$ -> $F_t^1$-> $F_t^2$->...-> $F_t^N$ ->$p_t(x)$. For simplicity, we denote $x$ as $F_t^0$, and $p_t(x)$ as $F_t^{N+1}$.Denote $I$ as mutual information and $H$ as entropy, as the process is deterministic,  items on the left of the chain knows all the information about those on the right,   i.e. $I(F_t^i;F_t^j) = H(F_t^j) (i<=j)$ , and the entropy of the features decreases as propagation proceeds, i.e. $H(F_t^0)$ > $H(F_t^1)$> $H(F_t^2)$>...> $H(F_t^N)$ >$H(F_t^{N+1})$ . Therefore, the feed forward process of a neural network gradually eliminates useless information, while retains information useful for obtaining $p_t(x)$  [1].
>
> When make a student feature $F_s^i$ to mimic a teacher feature $F_t^i$ w.r.t. L2 distance, we are actually maximizing $I(F_s^i; F_t^i)$, the mutual information between them , i.e. let $F_s^i$ know more about $F_t^i$  [2]. However, there are both task-relevant information and task-irrelevant information within $F_t^i$, and it is unknown that within  $I(F_s^i ; F_t^i)$, how much information are actually useful for determining $p_t(x)$ and how much will be discarded be lateral teacher modules.
>
> It is reasonable to assume that the student will benefit more from learning task-relevant information rather than task-irrelevant information, so we would *expect* that within $I(F_s^i; F_t^i)$, there are mostly task-relevant information. Formally, we hope the triplet mutual information,  $I(F_s^i; F_t^i; p_t(x))$, to be as large as possible, ( since $F_t^i$ completely determines $p_t(x)$, the triplet mutual information terms is actually identical to  $I(F_s^i; p_t(x))$  ). However, the loss $L2(B(F_s^i); F_t^i)$ cannot differentiate task relevant and task-irrelevant information, so our expectation cannot be realized.
>
> By feeding the transformed student feature $B(F_s^i)$ and the target teacher feature $F_t^i$ together into the lateral teacher modules, we are requiring that $F_s^i$ not only knows a lot about the original $F_t^i$ (i.e.  large $I(F_s^i; F_t^i)$), but also knows a lot about $M_t^{l} \circ \cdots \circ M_t^{i+1}\left(F_t^i\right)$ , where a lot of task-irrelevant information has already been erased. To meet such requirement, $B(F_s^i)$ has to learn more task-relevant information from $F_t^i$, which will *not* be discarded by lateral teacher modules.  In conclusion, from this perspective, FCFD is adding extra preference over task-relevant information to the knowledge that the student learns from the teacher.
>
> [1] Opening the black box of Deep Neural Networks via Information. arXiv:1703.00810, 2017.
>
> [2] Variational information distillation for knowledge transfer CVPR 2019.

---

> ### Author Response · Authors · 2022-11-11
> **Other problems**
>
> *Thank you for your comments and questions!*
>
> # Details of the Bridge Module B()
>
> When we say "with a bridge module $B_{st}^k$, which is a simple combination of convolution and BatchNorm layers", we mean $B(\cdot)$ is composed of exactly **one** convolution layer and **one** BatchNorm Layer. More Specifically, our implementation of the bridge module follows the implementation in the RepDistiller[1] github repo, which is the official repo of the CRD[2] paper. The kernel size is 3x3, and is of stride 2 when the target feature is 2 times smaller, and is of stride 1 when the target is as large as the source feature. When the target is 2 times larger than the source,  transposed convolution with kernel size 4 and stride 2 is used.  **In the revision we have clarified these details in Appendix A.1**. Compared with works like MGD[3] and Review[4], our bridge module is very light and alleviates the training cost.
>
>
>
> # Experiments
>
> **About *Some important baselines are missing***
>
> The results without  $\mathcal{L}\_{kd}$  are now updated in Tab. 1 and Tab.2. As the results show, FCFD w/o  $\mathcal{L}_{kd}$ still clearly outperforms existing methods.  If you think there are other missing comparisons that should be supplemented, please let us know.
>
> In this paper, we strictly follow the popular public benchmark established by CRD[2] and compare our FCFD with existing works that also follow this benchmark.  We think this is the best way to make our comparisons convincing. Note that some existing works, while claim to follow this benchmark, changes some important settings like number of training epoch, teacher checkpoint (Methods in Tab1 & 2 are trained with pre-trained teacher checkpoints identical), selection of optimizer, etc. Still some other works only select few teacher-student pairs to report their results.  To ensure that our comparisons  are meaningful, such works are not included in tables. Moreover, some works may not exist in our comparison because they are not competitive.
>
>
>
> # Training cost v.s. Improvement
>
> We first note that the practical training speed of FCFD is actually **faster** than the widely-known KD methods Review [3] and CRD[2] . To train an epoch with resnet8x4-resnet32x4, FCFD spends 43.98s, while 48.40s and 54.02s for Review and CRD. FCFD also leads to less peak memory usage than Review. Following your suggestion, a detailed analysis of training cost is now updated in the revision (**Appendix F**). We know that the training cost of FCFD seems to be large at first glance, but the following points make the cost acceptable:
>
> 1. We only functionally match relatively deep features, so the *lateral part of the network* is not that large.
> 2. We use random sampling strategy to lower the overall training cost.
> 3. We use very light bridge module and do not incorporate extra modules for complex feature processing.
>
> Besides comparisons with SOTA KD methods over training speed and peak GPU memory usage, in Appendix F we also discuss some techniques that makes FCFD more efficient. We agree that this complexity analysis section is very important and we apologize for its absence in the last version.
>
>
>
> **Moreover, considering the overall developing speed of the KD community, we believe the improvement brought by FCFD *is* significant**. Considering the res34-res18 pair in Tab.2. CRD was proposed in ICLR2020 and its accuracy is 71.17\%. Years have passed and the current SOTA DKD[5] and SRRL[6]  raise the top-1 accuracy by 0.53% and 0.56\%, respectively. Meanwhile, our FCFD outperforms CRD by 1.08\%, and outperforms SOTA by 0.52%.  Due to some inherent features of the KD problem (e.g. fixed student architecture, augmentation, etc.), it is hard for a KD method to achieve improvements as significant as methods in areas like model architecture design.
>
>
>
> [1] https://github.com/HobbitLong/RepDistiller
>
> [2] Contrastive representation distillation, ICLR 2020
>
> [3] Masked Generative Distillation. arXiv preprint arXiv:2205.01529 (2022)
>
> [4] Distilling knowledge via knowledge review, CVPR 2021.
>
> [5] Decoupled Knowledge Distillation, CVPR2022
>
> [6] Knowledge distillation via softmax regression representation learning, ICLR 2021

---

> ### Author Response · Authors · 2022-11-17
> **Sincerely expect your further comments**
>
> Dear Reviewer ScN2,
>
> Considering the deadline for discussion stage 1 is approaching, we sincerely ask if our response has addressed your concerns. If not, could you please list your follow-up questions? We are happy to have a further discussion with you. Thanks!
>
> Bests, Paper311 authors

---

> > ### Comment · Reviewer_ScN2 · 2022-12-10
> > **my final comments**
> >
> > I carefully read the rebuttals and still have several concerns.
> >
> > 1. Experiment:
> > The results w/o $\mathcal{L}_kd$ is counter-intuitive.
> > In table 1 where teacher and student are of similar architectures, the average of FCFD (proposed method) is $75.02\%$ while the FCFD w $\mathcal{L}_kd$ is $75.13\%$.
> > In table 1 where teacher and student are of different architectures, the average of FCFD (proposed method) is \textbold{$75.17\%$} while the FCFD w $\mathcal{L}_kd$ is \textbold{$75.18\%$}.
> > Since the results are so close, I believe FCFD does not provide orthogonal contributions.
> > What's more, in most cases FCFD performs even better than FCFD w $\mathcal{L}_kd$, which is counter-intuitive.
> >
> > 2. Training cost:
> > The author still does not address my concern about training cost.
> > Since the proposed method adopts random sampling strategy to lower the overall training cost, people can also use the similar strategy or modify the hyper-parameters of CRD and Review such as using less negative samples and reducing the number of connection layers in Review.
> >
> >
> > 3. Mathematically Insight:
> > Mutual information takes joint distribution into consideration. For instance,  we sometimes regard contrastive learning as a form of mutual information because large amounts of negative samples are leveraged to mimic the joint distribution.
> > Evidently, mutual information is not suitable to be used here as the mathematical insights behind FCFD.
> > On the other hand, what I really want to know is the mathematical insights on the negative aspects of l2 loss and the author does not answer it.
> >
> > 4. Novelty:
> > I also read other comments and found that some similar methods such as cross-distillation [1] and residual distillation [2] have already been proposed, which somehow limits the novelty of the proposed method.
> >
> > [1] Few Shot Network Compression via Cross Distillation, AAAI 2020.
> >
> > [2] Residual Distillation: Towards Portable Deep Neural Networks without Shortcuts, NeurIPS 2020.
> >
> > Overall, I keep the original score and do not recommend the acceptance of the paper.

---

### Decision · Program_Chairs · 2023-01-20

**Decision:**

Accept: poster

**Justification For Why Not Higher Score:**

Reviewers were split on this paper. After extensive discussion with the authors, most concerns were alleviated, but a few issues remained, including concerns about the novelty, the mathematical motivation, and the fairness of the experiments. I find these concerns to be valid and encourages the authors to continue to address them, but I feel that 1) the deficiencies of L2 are well described in the Section 1.3, and 2) the majority of the experiments are fair and additional improvements can be made for camera ready. The main remaining weakness is the novelty, which I agree is limited, hence I advocate accept at just the poster level.

**Justification For Why Not Lower Score:**

N/A

**Metareview: Summary, Strengths And Weaknesses:**

Summary:
This paper induces a new loss function and architecture for knowledge distillation. Prior work penalizes a distance metric between student and teacher features. This paper argues that instead a measure of "function (in)consistency" should be penalized. At layer L of the networks, features are extracted from student and teacher nets and sent to the latter layers (L+1 and beyond, called "lateral net") of the _opposite_ net -- teacher and student nets respectively. Then disagreement is penalized in the outputs. This encourages student features to be aligned with the teacher lateral net and vice versa.

Strengths:
* Reasonable idea with good motivation
* Improved performance on a variety of benchmarks

Weaknesses:
* Several prior works have used a similar "swapping" strategy, inputting student features into teacher lateral net (Cross Distillation, Residual Distillation, and SRRL)
* Some concerns about the fairness of the detection experiments
* Mathematical motivation was not convincing to all reviewers


**Note From Pc:**

if the above contains the word "oral" or "spotlight" please see: "oral" presentation means -> notable-top-5% and "spotlight" means -> notable-top-25%. As stated in our emails, we are disassociating presentation type from AC recommendations